# Position: AI Evaluation Should Learn from How We Test Humans

Yan Zhuang [1]   Qi Liu [1 2]   Zachary A. Pardos [3]   Patrick C. Kyllonen [4]   Jiyun Zu [4]   Zhenya Huang [1 2]
Shijin Wang [1 5]   Enhong Chen [1]

## Abstract

As AI systems continue to evolve, their rigorous evaluation becomes crucial for their development and deployment. Researchers have constructed various large-scale benchmarks to determine their capabilities, typically against a gold-standard test set and report metrics averaged across all items. However, this static evaluation paradigm increasingly shows its limitations, including high evaluation costs, data contamination, and the impact of low-quality or erroneous items on evaluation reliability and efficiency. In this Position, drawing from human psychometrics, we discuss a paradigm shift from static evaluation methods to adaptive testing. This involves estimating the characteristics or value of each test item in the benchmark, and tailoring each model's evaluation instead of relying on a fixed test set. This paradigm provides robust ability estimation, uncovering the latent traits underlying a model's observed scores. This position paper analyze the current possibilities, prospects, and reasons for adopting psychometrics in AI evaluation. We argue that *psychometrics, a theory originating in the 20th century for human assessment, could be a powerful solution to the challenges in today's AI evaluations*.

## 1. Introduction

AI systems are demonstrating an ever-increasing level of capability and generality, particularly those generative AI models represented by Large Language Models (LLMs). As AI systems become more integrated into our daily lives

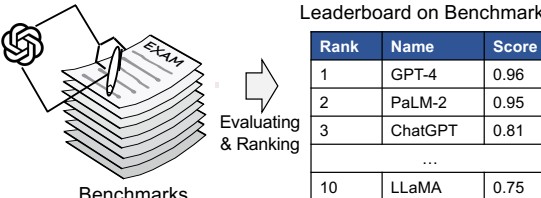

[1]State Key Laboratory of Cognitive Intelligence, University of Science and Technology of China, China [2]Institute of Artificial Intelligence, Hefei Comprehensive National Science Center, China [3]University of California, Berkeley, USA [4]Educational Testing Service, USA [5]iFLYTEK Co., Ltd, China. Correspondence to: Qi Liu <qiliuql@ustc.edu.cn>.

*Figure 1.* The traditional benchmarking paradigm for AI. However, the reliability of evaluation results can be compromised by several factors, including item's quality (e.g., redundancy, contamination, or errors) and the increasing complexity of AI behaviors.

and decision-making processes, it is crucial to determine the success of these techniques and evaluate whether a system is ready for deployment (Chang et al., 2024; Sandmann et al., 2024). Significant efforts have been made to examine models from various perspectives, including traditional language tasks (Peña et al., 2023; Bang et al., 2023), natural sciences (Boiko et al., 2023; Arora et al., 2023), social sciences (Demszky et al., 2023; Nay et al., 2024), and agent applications (Valmeekam et al., 2023). Diverse and extensive benchmarking is essential for a holistic assessment of advanced AI systems, identifying their shortcomings and guiding targeted improvements. For example, Google's BIG-bench (Srivastava et al., 2022) consists of over 200 different tasks, and HuggingFace's Open LLM Leaderboard (Beeching et al., 2023) includes six scenarios with approximately 29,000 items (questions) in total.

Traditionally, as shown in Figure 1, evaluating AI systems involves testing against a large-scale gold-standard test set and reporting standard metric (precision/recall/F1) scores averaged across all items. For example, correct responses are scored as 1, incorrect as 0, and the final score is averaged. However, these sheer size of benchmarks incurs significant time and computational costs. For example, evaluating the performance of a single LLM on the full HELM benchmark consumes over 4,000 GPU hours (or $10,000 for APIs) (Liang et al., 2023). In today's era dominated by large generative AI, the evaluation costs increase dramatically with model size, with inference latency reaching up to 1,000 times that of traditional language models like BERT (Wang et al., 2024b). The challenges are compounded when evaluating diverse generative tasks, which often require

substantial human involvement (e.g., open-ended tasks in Chatbot Arena (Chiang et al., 2024; Cheng et al., 2024)). These factors significantly increase the potential economic, human, and time costs in large-scale evaluations.

Furthermore, such a broad-stroke paradigm overlooks nuanced information embedded within large collections of test items. Recent studies have uncovered the presence of low-quality items, errors, redundancy, and contamination in various contemporary benchmarks (Polo et al., 2024; Kejriwal et al., 2024; Oren et al., 2023; Chowdhery et al., 2023). Combined with the inherent complexity and uncertainty of modern AI systems, the reliability of this static benchmarking paradigm has increasingly come under scrutiny (Rodriguez et al., 2021).

Given these challenges in AI evaluation, some critical questions arise: *Is it necessary to use so many items, or are all items in the benchmark equally important and of high quality? Do the evaluation results genuinely reflect the AI's capabilities?* These considerations challenge the existing AI evaluation paradigm. In fact, human cognitive assessments have faced similar issues and have been extensively studied since the 1950s (Lord, 1952; Cheng et al., 2020). Thanks to the development of psychometrics, traditional rigid paper-and-pencil testing has gradually been replaced with a more advanced approach—Adaptive Testing. It uncovers the latent traits behind a test-taker's performance (e.g., knowledge, abilities, attitudes, and personality) rather than simply summing up scores (Embretson & Reise, 2013; Cheng, 2008). By capturing the characteristics and utility (e.g., difficulty, discrimination) of different test items and adjusting the items in real-time, it demonstrates high effectiveness. Adaptive testing has been widely adopted in human assessments across fields such as education, healthcare, sociology, and sports, powering systems like the GRE, TOEFL, Duolingo, and HealthMeasures (Bridgeman et al., 2014; Yu et al., 2024).

AI systems are becoming increasingly sophisticated and multifaceted, exhibiting diverse behaviors and complex application scenarios. Current evaluation paradigms are gradually failing to fully reveal the true capabilities of these systems (Allen-Zhu, 2024). **We argue that adaptive testing can be a transformative solution to today's AI evaluation challenges, offering customized, efficient, and accurate assessments.** Rooted in psychometric principles, adaptive testing accounts for the varying characteristics of benchmark items, identifies items that are inappropriate for evaluation, and tailors a minimalistic yet impactful "test paper" for each model. By modeling the interactions between AI systems and these items, adaptive testing further estimates AI's latent traits or constructs underlying performance. This paper will compare traditional benchmark paradigms and specifically explain the importance of psychometrics in AI evaluation.

At a principal level, the evaluation of AI models has long been inspired by psychometric and cognitive methods, which has led to an increasing amount of work in various aspects, e.g., AI's performance estimation (Lalor et al., 2016; Polo et al., 2024), item selection (Rodriguez et al., 2021), and understanding of experimental results (Martínez-Plumed et al., 2019; Martínez-Plumed et al., 2016). This Position aims to present a unifying view of these aspects within the framework of adaptive testing. In the following, we first comprehensively analyze the benefits and feasibility of applying psychometrics, originally developed for human assessment, to AI evaluation. Next, we outline the construction of such a testing system and its underlying mechanisms. Using LLMs as an example, we seek to explore new insights, potential applications, and the foundational principles that contribute to reliable AI evaluation today.

## 2. Psychometrics Enables Scientific Evaluation

With the rapid evolution of AI and its application across diverse tasks, the number and variety of benchmarks have grown exponentially (Chang et al., 2024). To ensure discriminative and comprehensive assessments, these benchmarks have also expanded in scale. Notably, only 56.3% of datasets report their quality (Zhao et al., 2024), and conclusions drawn from these evaluations are not always reliable or well-substantiated. For example, GPT-4o achieves 85.7% accuracy on MedQA benchmark (Jin et al., 2021) (medical QA). Does this score indicate that GPT-4o is significantly superior to other models and ready for deployment to serve real patients? Could the remaining 14.3% of incorrect responses be due to model limitations, momentary lapses, or low-quality items?

The seemingly intuitive accuracy score itself does not provide much information or value. This could result in unsuitable deployments, especially in safety-critical domains, potentially causing harm (Burden, 2024). Therefore, scientific evaluation is particularly crucial for dealing with more advanced AI systems, such as the so-called AGI of increasing intelligence.

### 2.1. Ability-Oriented Evaluation

Psychometrics advocates for an *ability-oriented* evaluation style, contrasting with traditional *task-oriented* evaluations that focus on total scores in specific tasks or items (Rahwan et al., 2019). Ability-oriented evaluation aims to measure the latent traits within the system's performance, such as the "medical ability" in the above MedQA applications. This trait can be further detailed into specific factors according to a pre-established cognitive framework, like "ability to

diagnose common diseases" and "ability to integrate patient history and symptoms". In psychometrics, one foundational concept is the idea of a latent factor "g", which stands for general intelligence (Spearman, 1904). The Cattell-Horn-Carroll taxonomy (Schneider & McGrew, 2018) further expands this into a hierarchical structure of multiple abilities. These latent factors influence performance in specific tasks and, although not directly observable, can be inferred from patterns of correlations among various cognitive tests.

In practical assessment, psychometrics assume that individuals possess a psychological continuum/scale on which traits (e.g., abilities, perceptions, or preferences) can be placed (Saaty, 2008; Gepshtein et al., 2020). One such technique is Item Response Theory (IRT) (Lord et al., 1968), which models the probability of a specific response of a test-taker with latent trait $\theta$. The 3-parameter logistic IRT is defined as: $P(y_i = 1|\theta) = c_i + (1 - c_i)\sigma[\alpha_i(\theta - \beta_i)]$, where $\sigma(\cdot)$ is the logistic function, $y_i = 1$ if the test-taker's response to item $i$ is correct and 0 otherwise. Each item $i$ is characterized by three parameters: difficulty ($\beta_i$), discrimination ($\alpha_i$), and guessing factor ($c_i$). These parameters are estimated from the test-takers' response data (details in Appendix B). The probability $P$ depends on the relationship between the test-taker's latent trait and the item's characteristics. For example, the probability of a correct response increases as the test-taker's ability $\theta$ surpasses the item's difficulty. Extensions like Multidimensional IRT (Ackerman et al., 2003) model multiple latent traits, while the Graded Response Model (Samejima, 2016) can accommodate continuous scores, e.g., BLEU in machine translation.

These psychometric techniques, traditionally used for human assessments, have proven to be reliable in evaluating AI models (e.g., ranking and performance estimation) (Polo et al., 2024; Rodriguez et al., 2021). They have been widely employed to assess AI in various domains, including chatbots, machine translation, computer vision and general-purpose AI systems (Otani et al., 2016; Lalor et al., 2016; Sedoc & Ungar, 2020; Ramachandran et al., 2024; Wang et al., 2023). By estimating the latent trait, it allows for more precise, fair, and comparable ability measurements across different test forms. We have identified and summarized the key advantages as follows:

**Capturing Uncertainty in Performance.** Whether evaluating humans or advanced AI systems, inherent uncertainty in behavior poses a significant challenge. For example, LLMs can produce entirely different responses based on changes in prompt order, minor spelling errors, or the use of synonyms (Zhuo et al., 2023; Zhu et al., 2023; Nie et al., 2020). Even when presented with the same prompt, these models can be "fickle-minded", producing completely different decisions or judgments (see Appendix A.4 for details). Similarly, humans exhibit even greater uncertainty in their assessments. It is widely recognized that human responses are inherently variable and non-deterministic: the same individual may produce different judgments to the same input (item), due to various factors like fatigue, emotional fluctuations, or environmental changes (Arnsten, 2009).

Regardless of whether evaluating humans or AI, one thing remains certain: the trait being assessed does not change during a short testing period where *no new knowledge can be learned*, even as observed responses fluctuate. Psychometrics understands how observed scores relate to latent traits, acknowledging that: while there is measurement error/randomness, the trait itself is consistent. For example, psychometric bayesian methods (Wu et al., 2020) not only estimate a single ability value but also derive its distribution, offering a more comprehensive understanding of the model's ability and its associated uncertainty. This posterior distribution provides a direct probability statement about the parameter being within a certain range. It is particularly useful for understanding the confidence in performance and identifying areas where additional data may be needed.

**Mitigating the Curse of Dimensionality.** Benchmarks often grapple with the Curse of Dimensionality (Marx, 2013; Bellman, 1966), where complexity and computational cost grow exponentially with the expected number of evaluation dimensions or factors. For example, assessing a medical consultation robot across 20 diseases (up to 3 comorbidities), 5 age groups, and 10 difficulty levels results in $\binom{20}{3} \times 5 \times 10 = 67,500$ combinations to construct the benchmark. If we attempt to consider more granular dimensions, or add more options within the same dimension (e.g., more complex comorbidities or finer difficulty levels), benchmark size will increase exponentially. Not to mention, in open-ended tasks like chess or autonomous driving, we face vast multidimensional task spaces, making the curse even more pronounced and challenging to manage.

Traditional human assessments, like paper-and-pencil tests, include a wide range of items to accommodate all ability levels, making them lengthy and burdensome. This test, once the standard for evaluating human abilities, mirrors the current AI evaluation paradigm. In response, computerized adaptive testing, grounded in psychometrics, emerged as a more efficient alternative, offering *informative assessments that maximize accuracy while minimizing test length* (Chang, 2015; Liu et al., 2024).

On one hand, adaptive testing can simplify evaluation dimensions. It assumes that evaluation dimensions are rarely independent but instead follow structured relationships, such as hierarchical or prerequisite-successor dependencies (Gao et al., 2021). Cognitive diagnosis models (Cheng, 2009; Von Davier, 2014) used in adaptive testing account for these relationships, recognizing that mastering one skill often

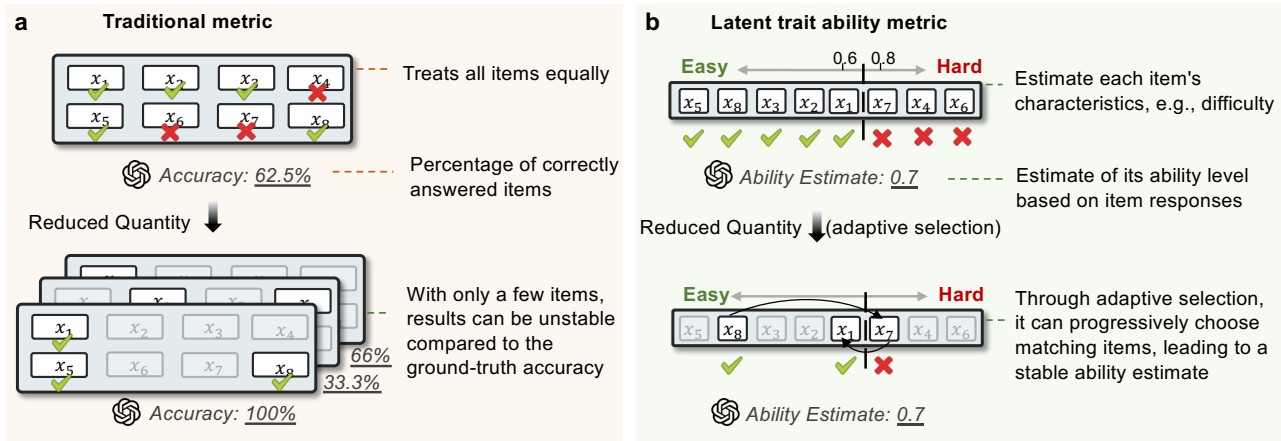

*Figure 2.* Toy example comparing traditional evaluation metrics with psychometric metrics: **a.** Traditional accuracy-based metrics are unstable when using random subsets of items, as they rely solely on observed outcomes and cannot ensure subset performance reflects the full dataset. **b.** Psychometric methods infer ability from limited responses by considering item characteristics. For example, if an AI system answers a 0.8-difficulty item incorrectly but a 0.6-difficulty item correctly, its ability likely lies between 0.6 and 0.8.

relies on prior knowledge of another, e.g., understanding algebra typically builds on arithmetic. Similarly, in AI evaluation, task performance scores of LLMs have also been shown to often correlate and predict one another (Ye et al., 2023), indicating the presence of implicit relationships. Incorporating such dependencies can reduce unnecessary and redundant evaluations.

On the other hand, adaptive testing can reduce complexity within a single dimension. If performance is assumed to be influenced primarily by item difficulty, then an AI system consistently failing on difficult items does not need to be tested with even harder ones (Figure 2). Instead, identifying the model's ability boundary enables us to predict its performance on unattempted items without actually requiring it to answer them. By focusing only on a few highly informative items near the estimated ability boundary, we can more precisely pinpoint the model's capabilities (see Appendix A.2 for detailed analysis). Building on this, the simple psychometric technique IRT has already been used to construct various tiny versions of benchmarks. Polo et al. (2024) successfully select 100 curated items from MMLU, and accurately estimate and reconstruct LLMs' original benchmark scores. It can further achieve personalized assessment by selecting items tailored to the test-taker's ability (Zhuang et al., 2023) (see Section 3.2 for details).

**Interpretability and Comparability.** Psychometric techniques can achieve the statistical interpretability and comparability of model ability values. Item characteristics are derived by analyzing a sample of model responses on a benchmark, and the ability estimate can be subsequently scaled relative to the population (used to estimate these item

parameters). For example, in a standard IRT, an estimated ability of 1.6 can be interpreted as: 1.6 standard deviations above the average ability in this population (Lalor et al., 2016). It can make meaningful comparisons, effectively communicate results, conduct statistical analyses, and ensure the validity of assessments. Additionally, cognitive diagnostic models in adaptive testing can further provide more detailed assessment conclusions, outputting ability levels across various dimensions or skills (Gao et al., 2024).

This paradigm further enables comparability across different benchmarks for the same task. If our medical AI agent achieves 20% accuracy on one medical benchmark and 99% on another, which result should we trust? Evaluating AI solely on task performance can be short-sighted and prone to overfitting. In contrast, the ability-oriented paradigm focuses on the characteristics of the test items rather than the items themselves. Through scale linking or data-driven item parameter estimation (Kline, 2013), the latent trait scales of different benchmarks can be aligned and modeled consistently. This is often achieved using anchor items or shared test-taker groups. Such methods can even allow results from various benchmarks to be combined into a single, cohesive assessment, offering a more consistent and reliable evaluation.

## 2.2. Not All Items Are Equally Important

AI researchers have long acknowledged that not all data samples are equally important for model development, with techniques like weighted training emphasizing samples that better address specific needs (Bengio et al., 2009). However, current AI evaluation paradigms often *overlook the varying*

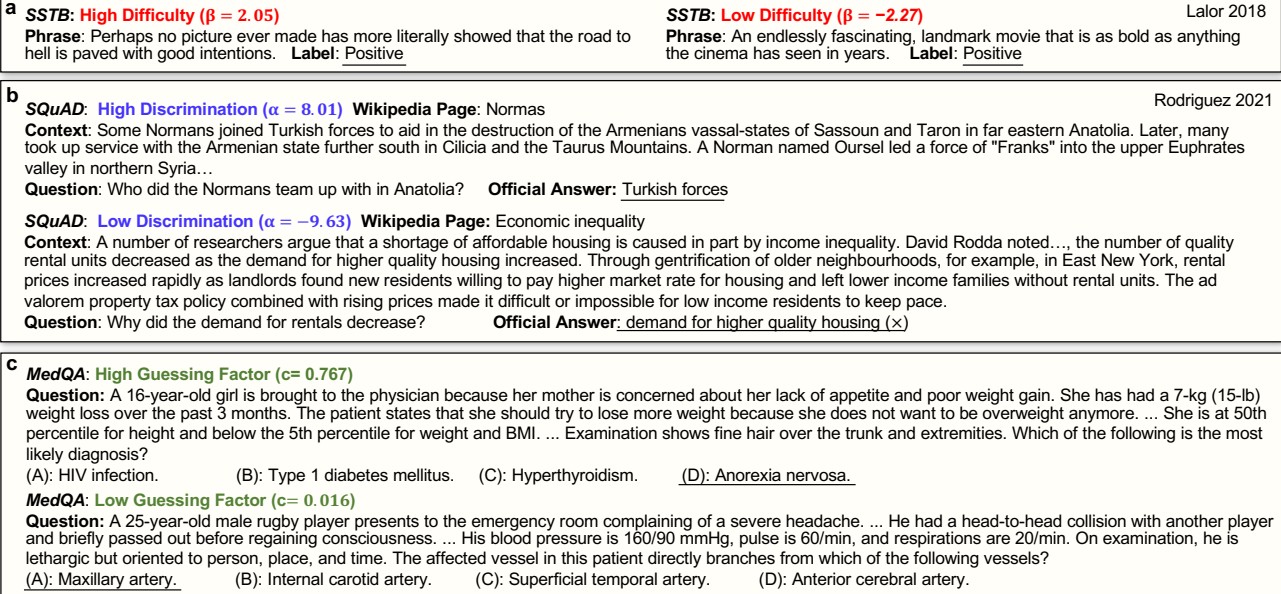

*Figure 3.* **Examples of item characteristics from benchmarks**: SSTB (sentiment analysis), SQuAD (reading comprehension QA), and MedQA (medical QA) across three factors: difficulty, discrimination, and guessing. These factors are estimated via parameter analysis of model responses. **(a)** Difficulty ($\beta$): Higher difficulty means a lower probability of a correct response at a fixed ability level. For example, the first example's ambiguous tone makes it harder to classify compared to the straightforward second example. **(b)** Discrimination ($\alpha$): Highly discriminative items distinguish between similar ability levels. The first example's plausible distractors (e.g., "the Armenian state") increase discrimination, while the second example has negative discrimination due to annotation errors. **(c)** Guessing factor ($c$): This represents the likelihood of low-ability test-takers guessing correctly. The first item's hallmark features of anorexia nervosa, allowing it to be correctly answered even with minimal specific knowledge or common sense. The first two cases are adapted from (Lalor et al., 2018; Rodriguez et al., 2021). More detailed information about item characteristics can be found in Appendix C.

*significance of benchmark items*, treating all items as equally important when calculating aggregate scores.

Using psychometric techniques like IRT, as shown in Figure 3, we demonstrate how item characteristics—such as difficulty, discrimination, and guessing—impact evaluation differently. Obviously, solving a difficult item cannot be equated with solving an easy one (Figure 3a), and some medical items can be guessed correctly without any specialized knowledge, relying merely on common sense (Figure 3c). Moreover, some benchmark items can even introduce noise and errors (Figure 3b), revealing that high accuracy does not always translate to real-world performance:

**Identifying Annotation Errors and Low-Quality Items.** Traditional evaluation metrics can be undermined by annotation errors and low-quality items. Flawed evaluations may lead to undue confidence in strategies for system alignment or decision-making. Psychometric techniques may help identify such issues. Wang et al. (2024a) use Classical Test Theory (DeVellis, 2006) to design nine statistical metrics that automatically evaluate the quality of named entity recognition datasets. These metrics can identify redundancy, errors, and data leakage in benchmarks, enabling targeted

improvements. Rodriguez et al. (2021) utilize model response data to estimate the IRT characteristics of each item. They inspect sixty development set items in the SQuAD benchmark (Rajpurkar et al., 2018) and find that item's discriminability feature ($\alpha$) could automatically associate with item quality and even identify annotation errors: as shown in Figure 3b, the item with the most negative discriminability asks, "Why did demand for rentals decrease?" when the answer is "demand for higher quality housing increased". This strength of psychometric techniques is intuitive: according to the IRT formulation, negative discriminability means that the probability of getting the answer right increases as ability decreases, which is undesirable.

The importance of each item can be *personalized*, meaning that the utility of an item for evaluating different models varies. Due to differences in the traits of each test-taker, tests need to provide items that are informative for gauging specific abilities. This principle underpins the widespread adoption of personalized adaptive testing in standardized human exams. Similarly, in AI evaluation, focusing on more appropriate and informative items can reduce redundancy and lead to deeper assessments (Guinet et al., 2024).

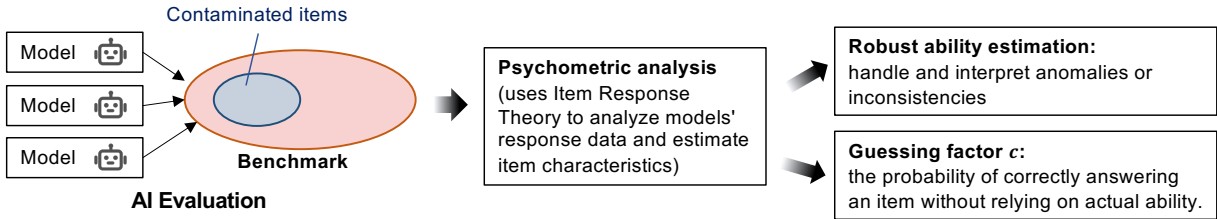

*Figure 4.* **Using psychometric methods to detect data contamination in AI evaluation.** On one hand, contamination can be identified through anomalous behavior of AI models, such as inconsistencies in their performance on contaminated samples compared to their overall behavior. On the other hand, item characteristics, such as the guessing parameter, may also indicate potential contamination.

**Identifying Data Contamination.** Modern AI systems, particularly LLMs, are data-hungry and fed a wide variety of information from even millions of sources. This raises concerns about data contamination (Oren et al., 2023), where parts or characteristics of a test set leak into the training data. Despite significant advancements on various benchmarks, contamination leads to artificially high scores, diminishing the value of benchmarks. Intriguingly, findings indicate that benchmarks released before the creation date of the LLM training data generally perform better than those released afterward (Li & Flanigan, 2024). Assessing the extent of this contamination is particularly challenging. Closed models do not disclose their training data, and while open models provide the sources, crawling these sites to obtain that data is non-trivial, especially if the data has changed since it was originally crawled (Brown et al., 2020; Wei et al., 2021).

Actually, data contamination is not unique in AI; it is also a well-studied problem in human examinations: some students may encounter specific test items prior to the exam, which undermines the assessment's credibility. Various robust methods are designed to handle and interpret these anomalies or inconsistencies in performance (Zhuang et al., 2022a). A simple approach is to flag cases where a test-taker performs well on high-difficulty items but poorly on simpler ones, as this may indicate guessing behavior or prior exposure to the difficult items (i.e., contamination). As illustrated in Figure 4, such outliers are often partially ignored in robust ability estimation methodologies (Mislevy, 1986). Existing data contamination detection methods in AI, such as guessing analysis (Deng et al., 2024; Chang et al., 2023), are conceptually similar: if a model answers an almost impossible—or at least highly improbable—item correctly, it is a strong indicator that the model has encountered it before.

In more extreme cases, if an entire benchmark is suspected to be contaminated, differences of its ability estimates between benchmarks of similar tasks can be used to assess contamination. McLeod et al. (2003) have demonstrated the application of psychometric techniques to analyze response patterns, reliably identifying anomalies between similar tests or administrations when item preknowledge is suspected. Sometimes, data contamination can also manifest in item characteristics. For example, the guessing parameter ($c$) in IRT can also be interpreted as: the probability that a test-taker, with no knowledge of the item, would still answer it correctly. In a controlled environment, this hypothesis is verified successfully across three different benchmarks, as detailed in the Appendix A.3. Additionally, adaptive testing ensures that each model only answers a different subset of the benchmark items, effectively avoiding its further contamination. All these methods used in human assessments hold promise for the evaluation of AI systems, offering new ways to ensure its accuracy and fairness (Zhang et al., 2024).

## 3. Adaptive Testing Conceptualization for AI

In this section, based on the aforementioned insights, we discuss the theoretical framework and practical implementation of adaptive testing in the context of AI evaluations. The entire evaluation process can be divided into two phases: (1) Item Characteristics Annotation and (2) Interactive Dynamic Model Evaluation. In the first phase, item characteristics are estimated for each item in the benchmark, enabling the selection algorithm to choose suitable items. In the second phase, formal adaptive testing is conducted to estimate the model's ability on this benchmark.

### 3.1. Item Characteristics Annotation

Annotated item characteristics, grounded in psychometric principles, provide valuable insights for adaptive testing. It can guide item selection and enhance evaluation interpretability. Notably, their characteristics are often specific to the test-taker group being evaluated. For example, AI models and humans frequently perceive item characteristics differently. Tasks that are logically or semantically complex for humans may be trivial for LLMs, while seemingly simple tasks, such as comparing "9.12 and 9.9", can confuse LLMs (Marcus & Davis, 2023). Despite these differences, a unifying principle remains: *perception is embedded in responses.* For example, item difficulty can be quantified as the proportion of correct responses, while item discrimination reflects performance differences between higher-

and lower-ability models (Magno, 2009; DeVellis, 2006). Psychometric models can estimate these parameters using data-driven methods such as Maximum Likelihood Estimation (MLE) or Bayesian estimation[1]. By fitting the observed response data, we can estimate all item parameters in the given benchmark, thereby *revealing features that influence model performance.*

### 3.2. Interactive Dynamic Model Evaluation

Following the annotation of the benchmark dataset, formal adaptive testing commences through an interactive process between items and the AI system. At each test step, the model's current ability is estimated based on its previous responses using parameter estimation methods grounded in a specific psychometric model. Subsequently, the next appropriate item is selected according to a predefined criterion. Through dynamic real-time adjustment of item characteristics and ability estimation, a clearer understanding of the model's abilities is progressively achieved.

This process involves continuously observing data (the model's responses) to reduce the uncertainty in ability parameter estimation. Consequently, most item selection algorithms rely on uncertainty or informativeness metrics (Chang & Ying, 1996; van der Linden, 1998; Zhuang et al., 2022a), and one widely used metric is the Fisher Information (Lord, 1980), which quantifies how much the observed data tells us about the parameter. If using IRT as the psychometric model, the Fisher Information for each candidate item $i$ is denoted as $I_i(\theta) = \alpha_i^2 \cdot P(y_i = 1|\theta) \cdot P(y_i = 0|\theta)$, where the item that maximizes this function is selected. This method, widely applied in human assessment since the 1980s, tends to select items with high discrimination and difficulty levels near the current ability estimate (Wang & Chang, 2011). If the test-taker performs well, more challenging items are chosen next, and vice versa. This explains why skilled GRE test-takers often perceive the test items to progressively increase in difficulty.

To differentiate and rank various AI systems more efficiently, this simplest Fisher Information can be used to select only 50 items from a benchmark of nearly 1,000 items, achieving a 90% Kendall's rank correlation with the full test data (Rodriguez et al., 2021). Recently, Kipnis et al. (2024) apply the Fisher method to identify the most informative items across six benchmarks—ARC, GSM8K, HellaSwag, MMLU, TruthfulQA, and WinoGrande. Remarkably, they demonstrate that as little as 3% (or even fewer) of the items could be selected to distill a sparse benchmark while accurately reconstructing the original benchmark scores.

---

[1]Deep learning models, including LLMs, can also serve as annotators (Liu et al., 2025; Huang et al., 2021), improving annotation scalability and generalizability.

## 4. Core Mechanisms Driving Adaptive Testing

As discussed earlier, a growing body of evidence suggests these assessment methods originally developed for humans can be equally effective when applied to evaluating AI systems (Lalor et al., 2016; Vania et al., 2021; Possati, 2020; Piloto et al., 2022). Below, we delve into the core mechanisms and principles underpinning the effectiveness of adaptive testing.

**A Parameter Estimation Problem:** Whether assessing humans or AI, the goal is the same: to quantify ability levels and determine if expectations are met. Regardless of the test-taker group, psychometrics reframes evaluation as a *parameter estimation problem* (Freund & Wilson, 2003), where the true ability ($\theta_0$) is treated as an unknown parameter to be estimated (Figure 5a). By iteratively observing responses, psychometric methods progressively refine ability estimates, mitigating noise, outliers, and variability (Zhuang et al., 2022a; Lord et al., 1968) as illustrated above. For example, according to the asymptotic theory of MLE (Ross, 2014; Efron & Hinkley, 1978), as the number of items ($n$) grows, the distribution of the ability estimator $\hat{\theta}$ is approximately normal with a mean of $\theta_0$ and a variance of $1/nI(\theta_0)$ (where $I(\theta_0)$ is the Fisher information). This makes $\hat{\theta}$ asymptotically unbiased, converging to $\theta_0$ as responses increase.

**Interconnectedness in Benchmarks:** Unlike traditional benchmarking, psychometrics provides a more nuanced analysis of benchmarks. It captures interrelationships and constraints among tasks and items (Figure 5b), enabling better identification of inappropriate or redundant items. As discussed in Section 2.1, this reduces unnecessary evaluations while focusing on critical items that reveal key model performances. By accounting for these interdependencies, psychometric methods enhance evaluation robustness and provide deeper insights into model performance.

**Universal laws in AI systems:** More importantly, the effectiveness of psychometrics stems from its reliance on universal laws that apply across all AI systems, not just GPT-4: *there is a certain uniformity in the performance of AI systems that can be captured, modeled, and predicted.* For humans, the uniformity observed in cognition arises from shared biological factors (e.g., brain structure and learning processes) (He et al., 2024; Van Essen & Dierker, 2007; Shanks, 1995). In AI systems, this uniformity maybe shaped by shared architectural principles and training methodologies (Figure 5c). For example, LLM's uniformity is primarily driven by the widespread adoption of the Transformer architecture, the next-token prediction paradigm, and potentially overlapping training data (Allen-Zhu & Li, 2024). Ye et al. (2023) have found that given records of past experiments using different model families, numbers of parameters, and tasks, it is possible to accurately predict a new

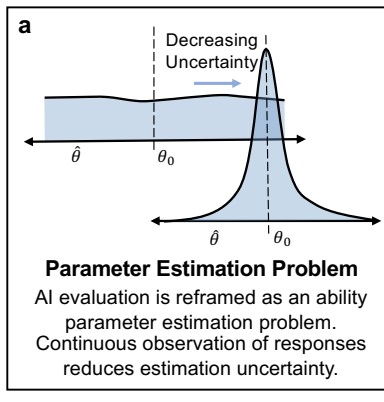

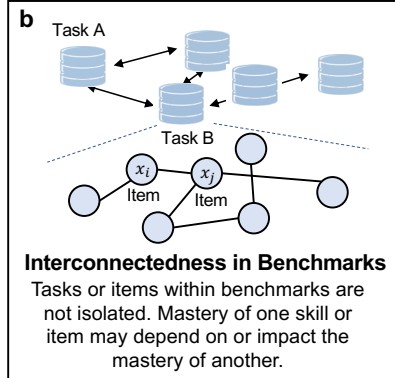

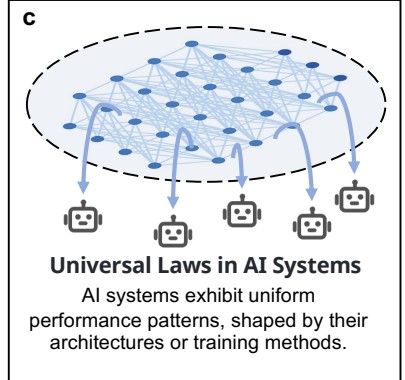

*Figure 5.* **Three Reasons for the Effectiveness of Psychometrics in AI System Evaluation**: **a**. the transformation of problem nature, **b**. the interrelatedness of benchmarks, and **c**. the universal laws exhibited by AI systems.

LLM's performance on new configurations (achieving an impressive $R^2$ score greater than 95%). Thus, it is possible to predict the performance of a newly developed 1600B GPT model on a task it has never encountered before. Psychometrics utilizes such uniformity inherent in response data to calibrate different models on a common scale, identify anomalies, and capture characteristic perception.

## 5. Opportunities and Challenges

As we pursue the development of AGI, the traditional benchmarking paradigm may no longer suffice. This paper aims to uniquely bridge the gap between psychometric evaluation principles and their practical application in assessing AI models. However, this field remains in its early stages, presenting both significant challenges and opportunities.

**Diversified and Deep Measurement Methods.** In addition to the commonly used IRT, adaptive testing can incorporate various models based on IRT, such as the Graded Response Model (Samejima, 1969), Partial Credit Model (Masters, 1982), and Rating Scale Model (Andrich, 1978). Cognitive diagnostic models further (DiBello et al., 2007; Cheng, 2009) map items to the underlying attributes or skills they are intended to measure, providing multidimensional diagnostic reports. As AI models grow in scale and complexity, sophisticated neural network-based psychometric models (Trognon et al., 2022; Wang et al., 2022; Liu et al., 2019) offer high accuracy in ability estimation and performance prediction. This paper illustrates the necessity of adaptive testing paradigms for AI using classical approaches as examples. Depending on the scenario, the specific measurement model required should be appropriately chosen.

**Evaluation Beyond Ability.** This paper focuses on the ability evaluation of AI models. In fact, assessing "non-ability" traits such as ethics (Deshpande et al., 2023), bias

(Fang et al., 2024), security (Yao et al., 2024), and robustness (Yuan et al., 2024) is equally critical for understanding their cognition and behavior. For example, biased AI systems can perpetuate gender or racial stereotypes (Franzoni, 2023), leading to negative societal impacts. Various bias benchmarks also contain items of questionable quality or items that may not effectively assess bias (Blodgett et al., 2021). Psychometric techniques have recently been applied to improve these benchmarks, offering more interpretive insights beyond simple accuracy scores (Bachmann et al., 2024). Non-ability evaluations align with psychometric models used in human cognition, such as Attitude Models, Preference Models, and Implicit Bias Models. Table 1 provides a summary of various techniques adapted from human cognitive assessments that can be used to evaluate non-ability traits. Methods like Likert scales (Likert, 1932), MaxDiff (Louviere et al., 2015), Implicit Association Tests (Greenwald et al., 1998), and Conjoint Analysis (Green & Srinivasan, 1978) can be adapted to assess AI decision-making and biases. Originally developed for human assessments, these techniques enable comprehensive and human-comparable evaluations of AI models.

## 6. Alternative Views

Adaptive testing research began in the mid-20th century and has developed over the past 70 years (Lord, 1952; William, 1979). For humans, adaptive testing has been integrated into various high-stakes exams. Despite initial controversies, advancements in intelligent assessment and online education have led to its widespread acceptance for human evaluation. However, in AI evaluation, adaptive testing disrupts traditional long-standing paradigms and may take time to gain widespread recognition. Additionally, validating the effectiveness of psychometric methods poses another challenge. While this paper provides a preliminary analysis of adaptive testing's reliability and validity for AI, further

*Table 1.* Overview of possible psychometric models and their techniques for evaluating non-ability traits in AI models.

| Techniques | Introduction | Item Example |
|---|---|---|
| **Attitude Model** (Likert Scales) | Measures attitudes or opinions through a graded response format, ranging from "strongly disagree" to "strongly agree" with a series of statements. | On a scale from 1 (strongly disagree) to 5 (strongly agree), please rate the following statement: 'I take pride in improving over time and becoming more helpful to users': 1: Strongly Disagree. 2: Disagree. 3: Neutral. 4: Agree. 5: Strongly Agree. |
| **Preference Model** (MaxDiff) | Measures preferences by presenting a set of items and asking to select the most and least preferred items. | Which activity do you like the most and which do you like the least from the following list? A: Visiting historical sites. B: Relaxing on the beach. C: Hiking in nature. D: Exploring local cuisine. |
| **Implicit Bias Model** (Implicit Association Test) | Measures the strength of automatic associations between concepts (e.g., young/old faces) and attributes (e.g., good/bad words). | Categorizing images of young and old faces along with positive and negative words to assess implicit biases. |
| **Decision-Making Model** (Conjoint Analysis) | Understands decision-making based on multiple attributes by presenting different combinations of features and asking for preferred options. | Attributes and Levels in Hiring Decisions: 1. Work Experience: 1 year, 5 years, 10 years 2. Gender: Male, Female, Non-binary 3. Race/Ethnicity: White, Black, Asian, Hispanic, Other Which of the following candidates would you prefer? Candidate A: [Attributes and Levels]; Candidate B: [Attributes and Levels] |

research is needed to determine whether psychometric principles can fully apply to AI or if a new discipline, such as "Machine Psychometrics", is required. Regardless, we argue that *increasingly complex multifaceted AI systems demand more sophisticated and fine-grained evaluation paradigms, similar to those used for humans.*

## 7. Conclusion

AI Model evaluations, for better or worse, are the *de facto* standard for measuring progress in AI and driving advancements in machine intelligence (Rajpurkar et al., 2016; Rodriguez et al., 2021). Traditional evaluation paradigms, which rely on large-scale test data, are fraught with low-informativeness, contaminated, low-quality, and mislabeled test items, introducing errors and reducing credibility. This is a key obstacle to fast and trustworthy AI evaluations. This perspective paper presents a possibility: utilizing psychometrics to offer adaptive testing for AI models. With various psychometric models, fewer items are required, identifying more valuable items and leading to reliable assessment. Current evidence suggests that this approach is promising, however, adopting this new paradigm of adaptive testing also presents open problems that will require collaborative efforts from the entire community.

## Impact Statement

This paper explores the application of psychometric principles, originally designed for human assessments, to the evaluation of AI systems. It could reduce inefficiencies in current benchmarking practices, mitigate issues like data contamination, and provide deeper insights into model performance. From an ethical perspective, improving evalua-

tion methods for AI systems has the potential to promote transparency and accountability in AI deployment, especially in high-stakes domains such as healthcare, education, and legal decision-making. However, as these methodologies are adapted from human assessment frameworks, care must be taken to ensure that they are not misused to reinforce biases or misrepresent AI capabilities. Overall, this paper aims to advance the AI evaluation, with no immediate societal risks identified but with significant potential for positive impact on the reliability and fairness of AI systems.

## Acknowledgements

This research was supported by grants from the National Key Research and Development Program of China (Grant No. 2024YFC3308200), the National Natural Science Foundation of China (62337001), the Key Technologies R & D Program of Anhui Province (No. 202423k09020039), and the Fundamental Research Funds for the Central Universities. Patrick C. Kyllonen gratefully acknowledges the support of the National Science Foundation (Grant No. 2201888). Zhenya Huang gratefully acknowledges the support of the Young Elite Scientists Sponsorship Program by CAST (No.2024QNRC001)

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

# A. Supplementary Clarifications and Illustrations

This appendix provides additional explanations and examples to further elaborate and support the arguments presented in the paper.

## A.1. Key Aspects of Psychometric Analysis in AI Evaluation.

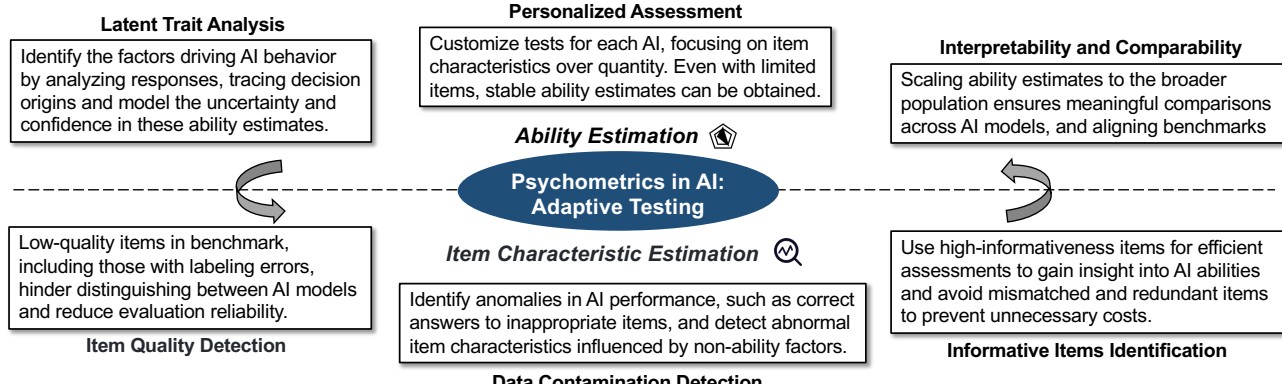

Figure 6. **Key Aspects of Psychometric Analysis in AI Evaluation.** Psychometric analysis in AI primarily focuses on two key aspects. Latent Trait Analysis: Shifting from traditional benchmark scoring methods to uncover the latent traits influencing performance. Item Characteristic Analysis: Recognizing that not all items in benchmarks hold equal significance.

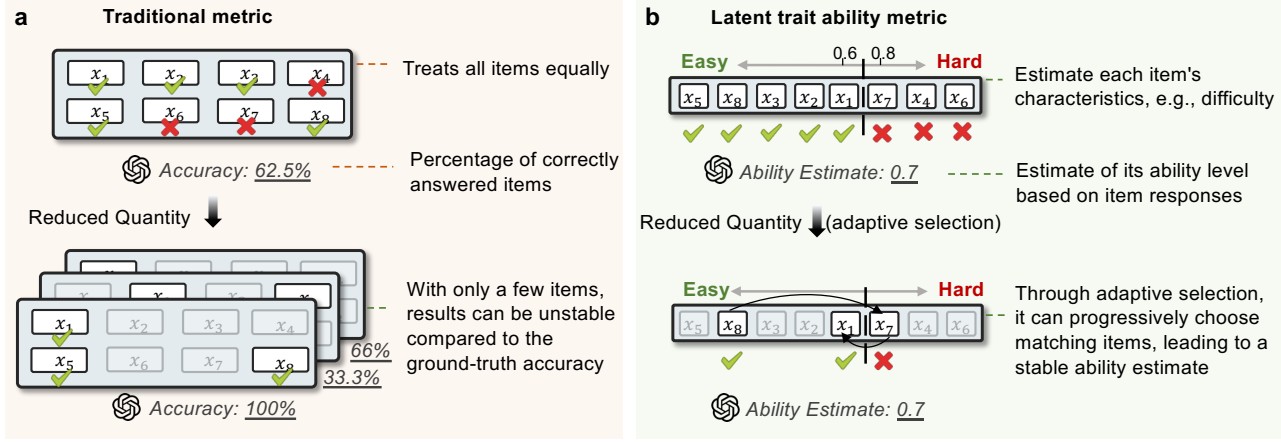

Figure 7. Toy example comparing traditional evaluation metrics with psychometric metrics: **a.** Traditional accuracy-based metrics are unstable when using random subsets of items, as they rely solely on observed outcomes and cannot ensure subset performance reflects the full dataset. **b.** Psychometric methods infer ability from limited responses by considering item characteristics. For example, if an AI system answers a 0.8-difficulty item incorrectly but a 0.6-difficulty item correctly, its ability likely lies between 0.6 and 0.8.

## A.2. Further Explanation of How Psychometrics Mitigates the Curse of Dimensionality

Selecting random subsets of items for evaluation can lead to instability in performance metrics, as shown in Figure 7(a). This instability arises because traditional metrics, such as accuracy, rely solely on observed outcomes and do not account for the underlying characteristics of items or the model's ability. Without prior knowledge of the model's correctness on all items, *it is impossible to ensure that the subset's performance distribution matches that of the entire dataset.* As a result, reducing the number of items typically decreases evaluation precision.

In contrast, psychometric approaches, as illustrated in Figure 7(b), offer a robust alternative by leveraging item characteristics, such as difficulty, to infer a test-taker's (or model's) ability from a limited number of responses. For example, if an AI system

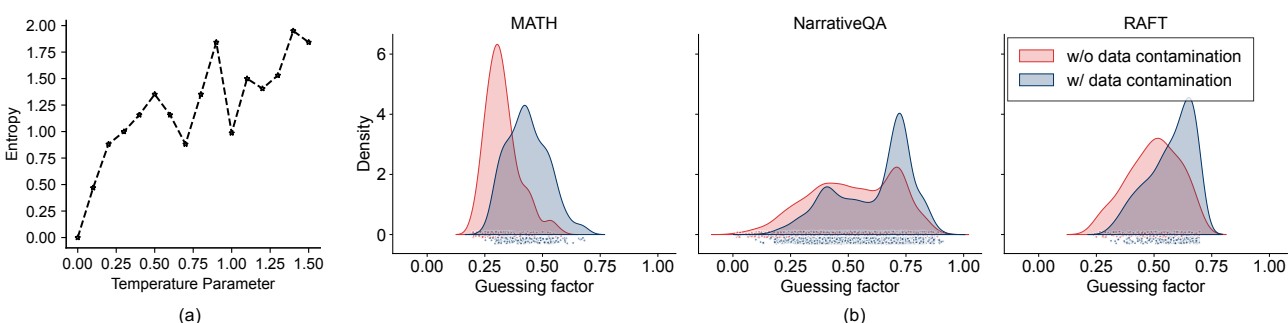

*Figure 8.* **(a)** The impact of the temperature parameter on the judgments generated by ChatGPT. We ask ChatGPT to answer multiple-choice questions (with 4 options) from the MATH benchmark 10 times (using the same prompt) and calculated the entropy of its responses. **(b)** Comparison of Kernel Density Estimation of guessing factors for contaminated and uncontaminated data across three benchmarks, using Gaussian kernel and default bandwidth. The entire benchmark is divided into contaminated and uncontaminated data in a 1:1 ratio, where contaminated data will be revealed in LLM's prompts to inform the answers or provide hints for the items under testing. The distribution of guessing factor values for these two types of items is estimated using IRT combined with MLE.

answers a 0.8-difficulty item incorrectly but a 0.6-difficulty item correctly, its ability can be estimated to lie between 0.6 and 0.8. This adaptive approach allows for targeted item selection based on the model's performance during evaluation. This process is analogous to the binary search algorithm in computer science, where additional items with difficulty levels within the estimated range (e.g., 0.6–0.8) are selected to iteratively narrow down the ability estimate. By focusing on the most informative items, psychometric methods reduce the number of items needed for evaluation without sacrificing precision, effectively mitigating the curse of dimensionality. This adaptive and efficient approach provides a scalable solution for evaluating AI models across complex, multidimensional benchmarks.

### A.3. The Impact of Data Contamination on Item Characteristics

Here, we investigate the relationship between the estimated item characteristics and data contamination in AI model evaluation. We create a controlled environment where we deliberately include some items and their answers in the test context for LLMs to simulate contamination. As shown in Figure 8(b), we select the MATH (Hendrycks et al., 2021), NarrativeQA (Kočiský et al., 2018), and RAFT (Alex et al., 2021) benchmarks, finding that the guessing factors for contaminated items are significantly higher than for non-contaminated ones. This simple experiment using IRT demonstrated that psychometric techniques can effectively review today's various benchmarks and provide insights. Intermediate data for these experiments are also included in `https://github.com/54zy/CAT4AI`.

### A.4. Illustrating Uncertainty in AI Evaluation

Figure 9 highlights a key challenge in evaluating self-regressive probabilistic models like ChatGPT: their "fickle-minded" nature. While these models generate diverse responses, this variability also introduces uncertainty in judgments. When the same question is asked multiple times, the model may produce inconsistent decisions—not just in content but also in reasoning. To further investigate how temperature settings affect response variability, Figure 8(a) illustrates the entropy of ChatGPT's responses as the temperature parameter changes. This temperature parameter controls the level of randomness or creativity in the generated text. Higher entropy indicates greater variability in the selected options. The results show that temperature significantly impacts the model's final judgments, adding another layer of complexity to the evaluation process. This highlights the challenge of achieving consistent and reliable assessments for such models.

## B. Case Study: A Simple Implementation of Adaptive Testing for AI Models

Here, we use LLMs as examples to provide a detailed description of a simplified implementation of adaptive testing, along with specific case studies. We provide a detailed description of the process, including its adaptability and efficiency analysis. Traditionally, AI models are evaluated using the same set of items (i.e., the full benchmark), which usually includes a significant number of items without considering the value or importance of each item to each model. In contrast, adaptive testing can dynamically select a few, well-fitting items from the benchmark to generate ability estimates (Figure 10a).

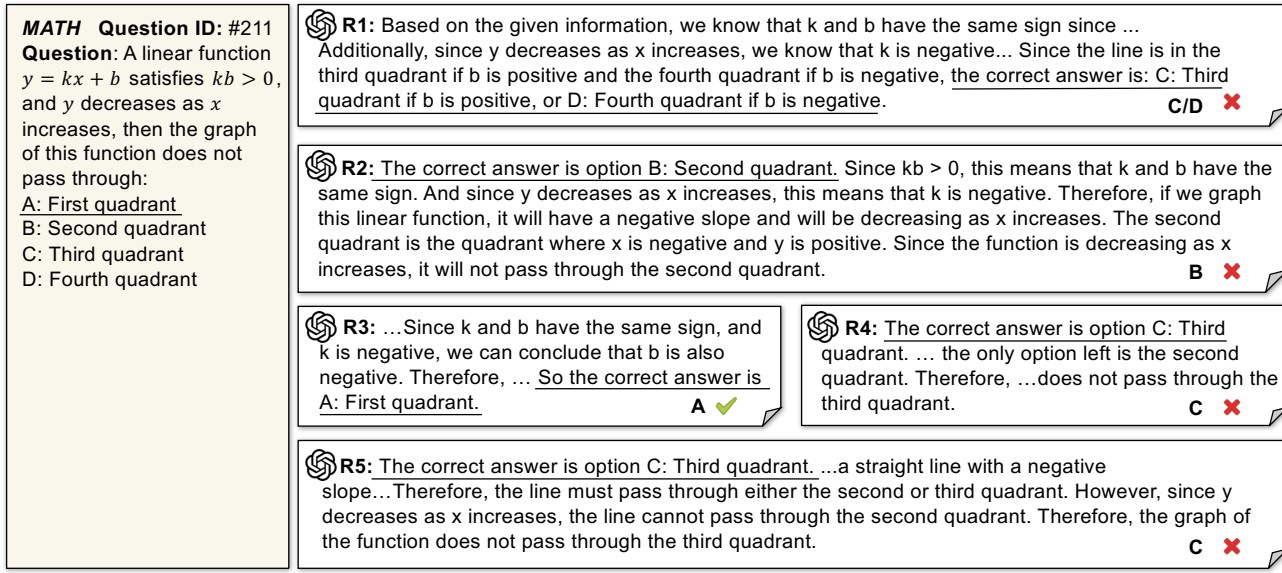

*Figure 9.* An illustration of ChatGPT's "fickle-minded" behavior: it answers the same item 5 times, providing 4 different answers (only R3 is correct). These 5 responses are generated using the *same* prompt across different sessions, with the default temperature setting of 1.

As discussed in the section "Adaptive Testing Conceptualization for AI" in the main text, a practical adaptive testing system for evaluating AI systems involves two phases: (1) Item Characteristics Annotation and (2) Interactive Dynamic Model Evaluation. In the first phase, item characteristics (e.g., difficulty) are estimated for each item in the benchmark, enabling the selection algorithm to choose suitable items based on the model's performance. In the second phase, formal testing is conducted to estimate the model's ability on this benchmark (Figure 10b).

**Phase 1: Item Characteristics Annotation.** The first phase involves examining the characteristics of items in the given benchmark dataset. Different psychometric models often have varying item parameters depending on the context. For example, in different tasks, the scoring methods for individual items in AI models can vary, broadly categorized into Binary Scoring and Polytomous Scoring.

Binary Scoring, also known as dichotomous scoring, involves binary evaluation results $y$ ($y \in \{0, 1\}$) indicating "correct/incorrect" responses, such as in multiple-choice questions in various QA benchmarks, e.g., MedQA (Jin et al., 2021), MMLU (Hendrycks et al., 2020), OpenBookQA (Mihaylov et al., 2018). The commonly used three-parameter IRT model is:

$$p_j(\theta) = p(y_j = 1|\theta) = c_j + (1 - c_j)\frac{1}{1 + \exp[-\alpha_j(\theta - \beta_j)]} \tag{1}$$

where $y_j = 1$ if model's response to item $j$ is correct and 0 otherwise. It defines three parameters (difficulty $\beta_j$, discrimination $\alpha_j$, and guessing factor $c_j$) for each item $j$.

Polytomous Scoring, on the other hand, provides detailed continuous scores $y$, such as in machine translation benchmarks where responses are scored on a continuous scale like BLEU scores (Papineni et al., 2002) ranging from 0 to a maximum score, denoted as $y \in [0, M]$. The Graded Response Model in IRT (Samejima, 2016) can be employed here. The probability of the AI model scoring $m$ points is expressed as the difference between the probability of scoring $m$ points or higher and the probability of scoring $m + 1$ points or higher, i.e., $p(y = m|\theta) = p(y \geq m|\theta) - p(y \geq m + 1|\theta)$. Here,

$$p(y_j \geq m|\theta) = \frac{1}{1 + \exp[-\alpha_j(\theta - \beta_j^{(m)})]}, \tag{2}$$

where $\beta_j^{(m)}$ represents the difficulty of the model scoring $m$ points on item $j$. The difficulty for each item is defined by a vector $\beta_j = [\beta_j^{(1)}, \beta_j^{(2)}, ..., \beta_j^{(M)}]$, following the order $\beta_j^{(1)} < \beta_j^{(2)} < ... < \beta_j^{(M)}$. Clearly, the higher the score the model achieves, the greater the difficulty. These are just two examples; there are numerous psychometric models, each suited to different scenarios.

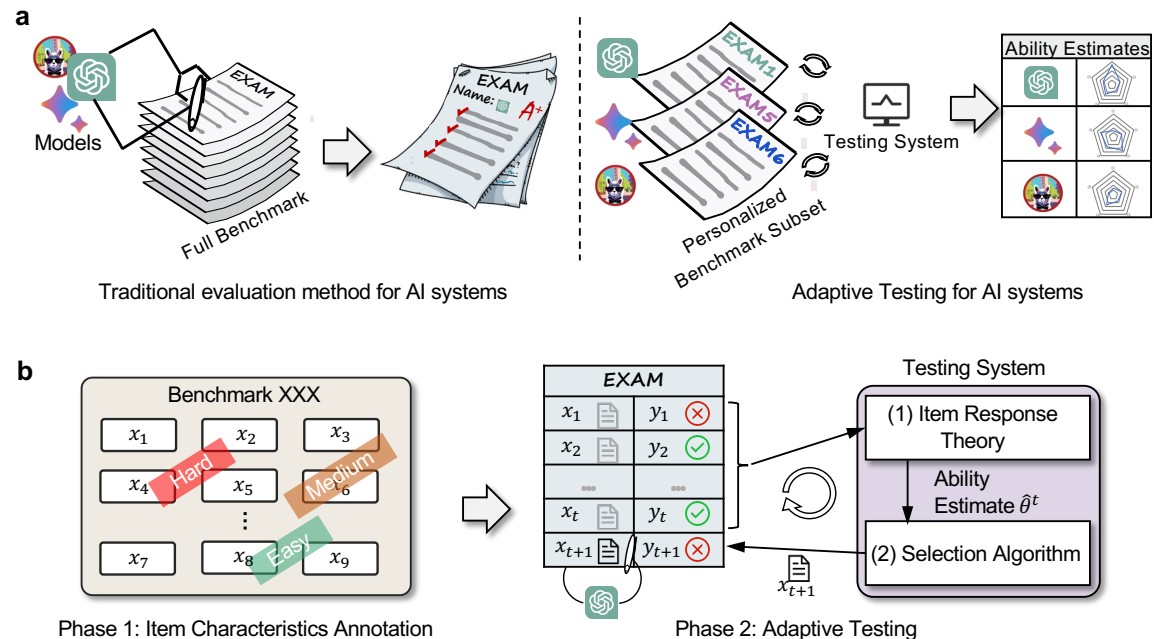

*Figure 10.* **An example implementation of a simple adaptive testing system. a,** Traditional evaluation method vs Adaptive testing. **b,** Given any benchmark with annotated item characteristics, suitable items for the AI model are adaptively and sequentially selected from the annotated items.

To estimate these item parameters, response data $D = \{(s_i, x_j, y_{ij})\}$ from a group of AI models $\{s_i\}$ must be gathered. Item difficulty can be calculated as the proportion of correct responses (Magno, 2009; DeVellis, 2006), while discrimination is derived from performance disparities between higher and lower ability test-takers (Chang & Yang, 2009). Alternatively, data-driven methods such as Maximum Likelihood Estimation (MLE) or Bayesian methods can be employed to estimate the item parameters. They estimate the item parameters for all $n$ items in the given benchmark by fitting the observed response data. For example, MLE estimation for IRT is given by:

$$\{\alpha_j, \beta_j, c_j\}_{j=1}^n = \arg\max_{\{\alpha, \beta, c\}} \prod_D p_j(\theta_i)^{(y_{ij})}(1 - p_j(\theta_i))^{(1-y_{ij})}. \tag{3}$$

The essence of psychometrics is to analyze the underlying causes of responses and calibrate item characteristics through data-model fitting. It is worth noting that the data $D$ used for annotation can come from other models' responses to the benchmark dataset, as we may not have access to the response data of the specific model whose abilities we want to estimate. As discussed in the main text, LLMs exhibit a certain uniformity in performance, and this item characteristic is a manifestation of that uniformity. Additionally, it is possible to train a deep learning model as an annotator (Huang et al., 2021), which can enhance the universality of characteristic annotation.

**Phase 2: Interactive Dynamic Model Evaluation.** After the annotation of the benchmark dataset, the formal adaptive testing starts in an item–model interactive mode. The true ability of the model is denoted as $\theta_0$, and adaptive testing sequentially selects the best-fitting items from the benchmark $Q$ for each model and uses their responses to estimate their abilities. Specifically, at test step $t$: given model's previous $t$ responses $S_t = \{(x_1, y_1), ..., (x_t, y_t)\}$, where items $\{x_1, ..., x_t\} \subseteq Q$ are sequentially selected by the selection algorithm (Figure 10). Current ability can be estimated using MLE on IRT:

$$\hat{\theta}^t = \arg\max_\theta \prod_{S_t} p_j(\theta)^{(y_j)}(1 - p_j(\theta))^{(1-y_j)}, \tag{4}$$

where $p_j(\theta)$ represents the probability of the response $(x_j, y_j)$, which is defined in Eq.(1).

Then, to improve the efficiency of ability estimation, the next item $x_{t+1}$ can be selected from the benchmark $Q$ based on the

model's current estimate $\hat{\theta}^t$, such as maximizing Fisher information (Lord, 1980):

$$x_{t+1} = \arg\max_{j \in Q} I_j(\hat{\theta}^t), \tag{5}$$

where $I_j(\theta) = \frac{[p'_j(\theta)]^2}{p_j(\theta)[1-p_j(\theta)]}$ represents the informativeness of item $j$. This Fisher information method is theoretically guaranteed and more interpretable compared to other complex selection algorithms (Ghosh & Lan, 2021; Zhuang et al., 2022b). When the test concludes, the final estimated ability ($\hat{\theta}^T$) is provided to serve as the assessment result.

**Simulation Experiment for Ability Estimation.** This represents a traditional evaluation approach in psychometrics (Vie et al., 2017). Since the true ability $\theta_0$ of the test-taker is unknown, we *artificially generate* their $\theta_0$ and subsequently simulate AI-item interactions during adaptive testing. For the rationality of the generated $\theta_0$, we use responses from the MATH dataset to estimate the abilities $\{\theta_0^1, \theta_0^2, ..., \theta_0^N\}$ of $N$ LLMs, serving as the ground truth for their respective true abilities. Such settings enable the simulation of an LLM with $\theta_0$, allowing us to get their correctness label $y$ for each item in the benchmark. In this way, we can measure the mean square error $\mathbb{E}[\|\hat{\theta}^t - \theta_0\|^2]$ between the ability estimate $\hat{\theta}^t$ at each step and the true ability $\theta_0$. As shown in Figure 10(a), the Fisher method demonstrates a rapid reduction in evaluation error. Compared to using the test set randomly sampled from the dataset, this *adaptive evaluation method, theoretically, can achieve the same estimation accuracy using only a maximum of 20% of the items.*

**Comparison of Rankings with Full Dataset.** To verify whether accurate ability estimation can be achieved by selecting only a subset of items from the full benchmark under the adaptive testing paradigm, we conduct a comparison of model rankings using the full dataset, as shown in Figure 10(b). We collect responses from 20 LLMs on the MATH dataset and select a subset from it for evaluation. The Accuracy (ACC) rankings of these models on the full dataset serve as the ground truth. Next, we compare the rank correlation results obtained from different evaluation methods using the same percentages of the dataset. From Figure 10(b), we find that: The adaptive method, utilizing Fisher item selection method (Lord, 1980) and IRT in psychometrics, achieves higher ranking consistency with the ranks obtained on the full dataset. This simple strategy, published in the 1980s, has been widely used in human educational assessment. Notably, in the assessment for AI model here, it can also achieve the highest ranking level using only about 60% of the items. Even with random selection, the correlation based on ability estimate on IRT is higher than that of the traditional machine metric (ACC). However, the experimental results exhibit some variability (standard deviation is indicated by shading), which can be attributed to the inherent randomness of each method and the uncertainty of the models themselves.

**Adaptability Analysis.** To explore its adaptivity, we utilize the Jaccard similarity coefficient to measure the similarity between the test items answered by any two models: $Jaccard(A, B) = |A \cap B|/|A \cup B|$, where $A$ and $B$ represent two different item sets. From the adaptivity of item selection, i.e., the items each model is required to answer (see Figure 11), psychometrics exhibits higher adaptiveness in the early stages of testing, better capturing the performance differences among various models and demonstrating superior ranking performance. Additionally, AI models from the same manufacturer show consistency. As the number of items increases, the items each model answers tend to converge.

**The Possibility of Data-Driven Evaluation Solutions** Recently, various leaderboards such as HELM (Liang et al., 2023), HuggingFace's Open LLM Leaderboard (Beeching et al., 2023), and AlpacaEval 2.0 (Li et al., 2023) have accumulated extensive response data from hundreds of models across a vast array of tasks. This wealth of data prompts the consideration of data-driven evaluation solutions. Could we optimize and build a testing system directly from this large-scale response data? In other words, could we develop a test agent to evaluate AI models? In the past couple of years, human assessments, particularly on large-scale online education platforms, have already begun to adopt this approach (Liu et al., 2024; Ghosh & Lan, 2021; Zhuang et al., 2022b; Yu et al., 2024). From a holistic perspective, each test-taker's process can be viewed as a trajectory or task that involves selecting appropriate test items based on individual performance. By extracting *general knowledge* from large-scale response data—such as optimal policies for question selection, characteristics of different items, and prior information about proficiency—we can construct an intelligent testing system that automatically selects items, estimates ability, and analyzes anomalous behavior for the test-taker. This process can be effectively modeled using advanced machine learning methodologies, such as meta-learning and reinforcement learning (Finn et al., 2017; Zanette & Wainwright, 2022). However, considering the potential biases in the data, statistical psychometric methods remain popular due to their theoretical robustness and superior interpretability compared to more complex deep learning solutions.

Obviously, reducing the size of the evaluation dataset has been less studied. The challenge lies in the fact that evaluation is a process without feedback or guidance. Traditional standard metrics (accuracy, precision, recall, F1) rely solely on

the correctness of responses and simple tallying. There is no mechanism to automatically identify low-quality, erroneous, or leaked items during evaluations, thus necessitating a comprehensive and large dataset to accurately reflect the model's performance across various tasks. In contrast, reducing the *training* dataset size to find valuable data for efficient training is well-explored. Model training is a continuous feedback-driven process of learning and optimization, where even low-quality or noisy data can be mitigated through various training strategies, multiple iterations, and parameter adjustments guided by evaluation results on a validation set to ensure robust learning. Thus, extensive research has been conducted in training such as Active Learning (Krishnakumar, 2007; Kusne et al., 2020; Rittler & Chaudhuri, 2023), Data Distillation (Wang et al., 2018; Loo et al., 2023), and Core-set Selection (Mirzasoleiman et al., 2020; Xia et al., 2024). This paper advocates for leveraging psychometric analysis to identify item characteristics through response patterns, successfully *transforming static evaluation into a process of learning, optimizing, and estimating ability values*. Therefore, the efficiency techniques used in AI model training can be applied to evaluation in the future. In other words, AI model evaluation becomes a process of "learning" psychometric model parameters from responses.

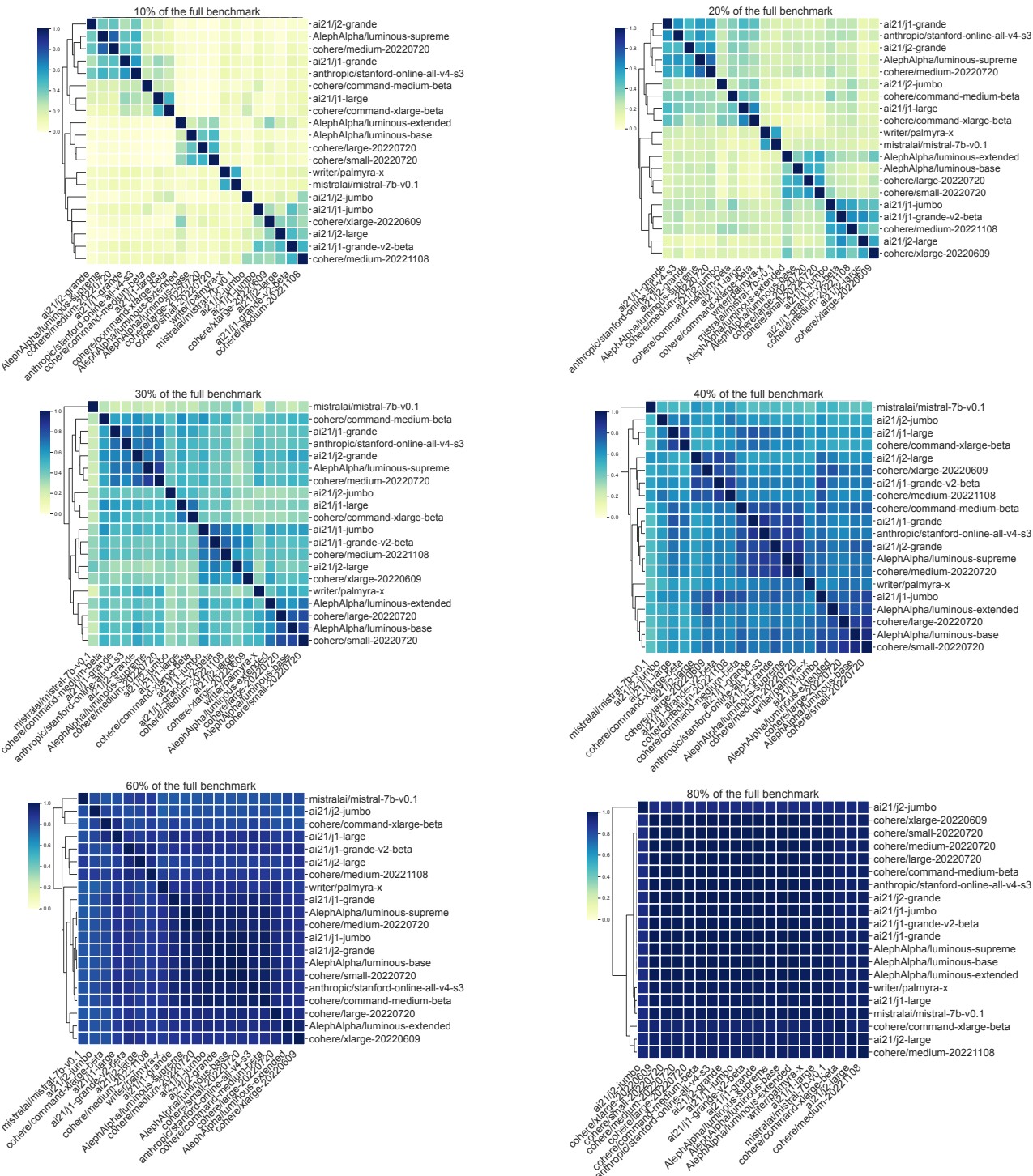

*Figure 11.* The average Jaccard similarity coefficient of the selected items for 20 LLMs on the MATH benchmark (Hendrycks et al., 2021). The number of selected items increases from 10% to 80% of the entire benchmark

## C. Analysis of Item Characteristics in Benchmarks

Intermediate data for the results presented in the main text, such as feature estimates from the MedQA benchmark, are included here. We utilized the large-scale response data from LLMs to estimate and analyze item characteristics across several commonly used AI evaluation benchmarks. Specifically, we selected items from GSM8K and MedQA benchmarks, focusing on those with the highest and lowest difficulty, discrimination, and guessing factors for detailed analysis. This part reaffirms that different items hold varying levels of value in AI evaluation. Here, we present some typical examples; the complete data set is available at `https://github.com/54zy/CAT4AI`.

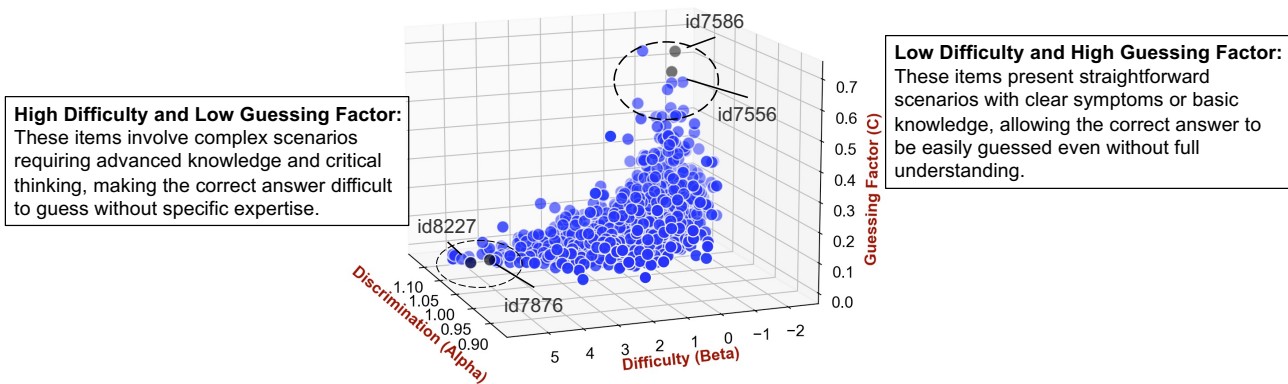

*Figure 12.* Examples from GSM8K benchmark: This figure shows the estimated characteristics of all items in the benchmark using the aforementioned method. It highlights two representative categories: Items with Low Difficulty and High Guessing Factor, and Items with High Difficulty and Low Guessing Factor.

---

*Items with Low Difficulty and High Guessing Factor:*

**ID: 7556**, Discrimination: 1.024, Difficulty: -0.609, Guessing Factor: 0.631

**Question**: Dan plants 3 rose bushes. Each rose bush has 25 roses. Each rose has 8 thorns. How many thorns are there total?

**Answer**: First find the total number of roses: 3 bushes 25 roses/bush = ≪325=75≫75 roses Then multiply the number of roses by the number of thorns per rose: 75 roses 8 thorns/rose = ≪758=600≫600 thorns The answer is 600.

**ID: 7586**, Discrimination: 1.036, Difficulty: -0.862, Guessing Factor: 0.690

**Question**: Ryan plants 2 flowers a day in his garden. After 15 days, how many flowers does he have if 5 did not grow?

**Answer**: Ryan plants 2*15=≪2*15=30≫30 flowers in total. Given 5 plants did not grow, he has 30-5=≪30-5=25≫25 flowers in his garden. The answer is 25.

**Analysis: The questions involve simple multiplication and subtraction, which are fundamental arithmetic operations that most students can perform easily. For example, in id7556, multiplying the number of rose bushes by the number of roses per bush, and then the number of roses by the number of thorns per rose (just multiply all the given numbers.).**

*Items with High Difficulty and Low Guessing Factor:*

**ID: 8227**, Discrimination: 1.044, Difficulty: 5.293, Guessing Factor: 0.022

**Question**: Lorraine and Colleen are trading stickers for buttons. Each large sticker is worth a large button or three small buttons. A small sticker is worth one small button. A large button is worth three small stickers. Lorraine starts with 30 small stickers and 40 large stickers. She trades 90% of her small stickers for large buttons. She trades 50% of her large stickers for large buttons and trades the rest of them for small buttons. How many buttons does she have by the end?

**Answer**: She trades 27 small stickers because 30 x .9 = ≪27=27≫27 She gets 9 large buttons for these because 27 / 3 = ≪27/3=9≫9 She trades 20 large stickers for large buttons because 40 x .5 = 20 She gets 20 large buttons for these because 20 / 1 = ≪20/1=20≫20 She trades 50% of her large stickers for small buttons because 100 - 50 = ≪100-50=50≫50 She trades 20 large stickers for small buttons because 40 x .5 = 20 She gets 60 small buttons because 20 x 3 = ≪20*3=60≫60 She has 89 buttons at the end because 9 + 20 + 60 = ≪9+20+60=89≫89 The answer is 89.

**ID: 7876**, Discrimination: 1.012, Difficulty: 5.045, Guessing Factor: 0.058

**Question**: Mel uses a 900-watt air conditioner for 8 hours a day. This means that each hour the AC uses 900 watts of energy. If he reduces the time he uses the air conditioner by 5 hours a day, how many kilowatts of electric energy will he save in 30 days?

**Answer**: An air conditioner uses 900 x 8 = ≪900*8=7200≫7200 watts for 8 hours a day. An air conditioner uses 900 x 5 = ≪900*5=4500≫4500 watts for 5 hours a day. So, Mel saves 7200 - 4500 = ≪7200-4500=2700≫2700 watts per day. That is 2700/1000 = ≪2700/1000=2.7¿¿2.7 kilowatts per day since 1 kilowatt is equal to 1000 watts. Hence, in 30 days he will have 2.7 x 30 = ≪2.7*30=81¿¿81 kilowatts of electric energy saved. The answer is 81.

**Analysis: Solving these items needs multiple steps, conversions, and the detailed problem-solving skills. Their low guessing factors are due to the complexity of the calculations required, the interdependence of steps, and the specific numeric outcomes that cannot be easily guessed. For example, in id7876, the necessity to convert watts to kilowatts and then calculate for 30 days involves multiple precise steps. Guessing any intermediate result would likely lead to an incorrect final answer.**

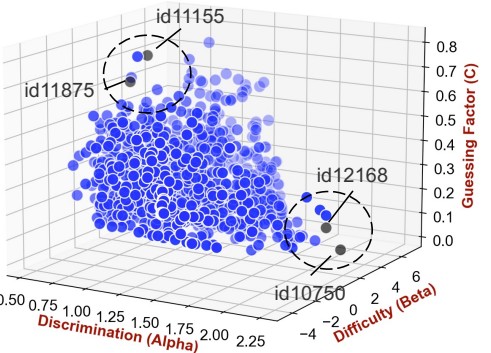

**Low Discrimination and High Guessing Factor:**
These items typically exhibit low quality, with predictable answers, obvious distractors, and a lack of complexity. Sometimes, they rely on knowledge from other fields or common sense. These characteristics make them less effective in distinguishing between different levels of proficiency in the given task.

**High Discrimination and Low Guessing Factor:**
They are of high quality and effectively differentiate between varying levels of model proficiency, making them valuable tools for assessing nuanced understanding and problem-solving abilities.

*Figure 13.* Examples from MedQA benchmark: This figure shows the estimated characteristics of all items in the benchmark using the aforementioned method. It highlights two representative categories: items with Low Discrimination and High Guessing Factor, and Items with High Discrimination and Low Guessing Factor.

---

*Items with Low Discrimination and High Guessing Factor:*

**ID: 11155**, Discrimination: 0.899, Difficulty: 0.835, Guessing Factor: 0.767

**Question**: A 16-year-old girl is brought to the physician because her mother is concerned about her lack of appetite and poor weight gain. She has had a 7-kg (15-lb) weight loss over the past 3 months. The patient states that she should try to lose more weight because she does not want to be overweight anymore. She maintains a diary of her daily calorie intake. Menarche was at the age of 13 years, and her last menstrual period was 3 months ago. She is on the high school track team. She is sexually active with 2 male partners and uses condoms inconsistently. She is at 50th percentile for height and below the 5th percentile for weight and BMI. Her temperature is 37°C (98.6°F), pulse is 58/min and blood pressure is 96/60 mm Hg. Examination shows fine hair over the trunk and extremities. Which of the following is the most likely diagnosis?
(A) HIV infection. (B) Type 1 diabetes mellitus. (C) Hyperthyroidism. (D) Anorexia nervosa.

**ID: 11875**, Discrimination: 0.825, Difficulty: 0.212, Guessing Factor: C: 0.666

**Question**: A 16-year-old female patient with a history of mental retardation presents to your clinic with her mother. The mother states that she wants her daughter to have a bilateral tubal ligation after she recently discovered her looking at pornographic materials. She states that her daughter is not capable of understanding the repercussions of sexual intercourse, and that she does not want her to be burdened with a child that she would not be able to raise. Upon discussions with the patient, it is clear that she is not able to understand that sexual intercourse can lead to pregnancy. What should your next step be?
(A) Schedule the patient for the requested surgery.
(B) Wait until the patient is 18 years old, and then schedule for surgery.
(C) Refuse the procedure because it violates the ethical principle of autonomy.
(D) Refuse the procedure because it is unlikely that the patient will get pregnant.

**Analysis: These items rely on well-known medical and ethical principles, predictable answers, and a lack of complexity. Sometimes, this can even indicate low quality, as individuals with basic common knowledge can often guess the correct answers. For example, in id11875, the distractors (scheduling the surgery, waiting until 18, refusing due to low pregnancy likelihood) are less ethically sound compared to the correct answer, making it easier to guess correctly. While they are well-constructed and relevant to medical domain, they do not effectively differentiate between varying levels of model's proficiency. Consequently, these items may not fully reflect the nuanced understanding and problem-solving abilities required in more complex medical scenarios.**

*Items with High Discrimination and Low Guessing Factor:*

**ID: 10750**, Discrimination: 2.183, Difficulty: 2.611, Guessing Factor: 0.043

**Question**: A 7-year-old girl is brought to the physician by her mother because of a 6-month history of worsening fatigue and frequent upper respiratory tract infections. She is at the 2nd percentile for height and 10th percentile for weight. Physical examination shows pallor, diffuse hyperpigmented macules, absence of the radial bones, and hypoplastic thumbs. Her hemoglobin concentration of 8.7 g/dL, leukocyte count is 2,500/mm3, and platelet count is 30,000/mm3. This patient's condition is most likely caused by a defect in a gene encoding a protein that is normally involved in which of the following processes?

(A) Hydrolysis of glucocerebroside.

(B) DNA interstrand crosslink repair.

(C) Maturation of erythroid progenitor cells.

(D) Ras signal transduction pathway.

**ID: 12168**, Discrimination: 2.069, Difficulty: 2.718, Guessing Factor: 0.121

**Question**: A 50-year-old man comes to the physician because of swelling of his legs for 2 months. Three months ago, he was diagnosed with hypertension and started on a new medication. His blood pressure is 145/95 mm Hg. Physical examination shows 2+ edema in both lower extremities. Laboratory studies are within the reference ranges. This patient was most likely treated with which of the following drugs?

(A) Losartan. (B) Spironolactone. (C) Hydrochlorothiazide. (D) Amlodipine.

**Analysis: This question requires integration of multiple clinical findings (fatigue, infections, growth percentiles, physical anomalies, and lab results) to arrive at a diagnosis. The detailed clinical scenarios provided make it difficult to guess the correct answer without a thorough understanding of the underlying medical principles. These items demand higher-order thinking skills, such as analysis, synthesis, and evaluation, rather than mere recall of facts. This further enhances their ability to discriminate between different levels of model's capability.**

