# OpenReview forum: "Position: AI Evaluation Should Learn from How We Test Humans"
_ICML.cc/2025/Position_Paper_Track — ICML 2025 Position Paper Track poster_

### Official Review · Reviewer_xfA5 · 2025-02-22

**Significance:** 3
**Argument Clarity:** 3
**Rating:** 4
**Confidence:** 4

**Questions:**

- Is "item characteristic annotation" aspect of the proposed framework used in picking the next question for evaluation at any given point? If yes, it would be great if authors can shed more light to it when they discuss the interactive dynamic aspect of the framework.
- On the flip side, when authors discuss the Interactive Dynamic Model Evaluation, in section 3.2., they end the subsection by mentioning "Recently, Kipnis et al. (2024) apply the Fisher method to identify the most informative items ... they demonstrate that as little as 3% (or even fewer) of the items could be selected to distill a sparse benchmark" which looks more relevant to item characteristics annotation rather than dynamically choosing the next question based on previous responses. Can the authors clarify?

**Discussion Potential:**

3

**Paper Summary:**

The paper advocates for using psychometric principles in evaluating AI systems performance. It lists challenges faced by traditional methods using large task oriented benchmarks to evaluate AI systems such as curse of dimensionality, low quality items, data contamination, etc. and explains how they are addressed in psychometric. Next, it proposes a two phase approach to evaluate AI systems that consists of item characterization annotation and interactive dynamic model evaluation. Authors argue that by dynamically selecting the next benchmark question based on the system's response to previous questions, a significantly smaller benchmark can effectively evaluate the system's performance and provide references for their claims.

## update after rebuttal
After reading authors' rebuttals and other reviewers' comments, I'd like to keep my overall "Accept" recommendation.

**Position:**

Yes

**Position In Title:**

Yes

**Related Work:**

3

**Strengths And Weaknesses:**

*Strengths*
- The topic, i.e. using psychometric principles to reduce the computational cost and increase the accuracy of AI system evaluation, is very relevant and is of great importance for the ICML community.
- The paper has done a good job providing references to the relevant psychometric literature and covering relevant recent efforts from the AI community.

*Weaknesses*
- No alternative view/position is presented in the paper.
- The proposed adaptive testing framework does not clearly address the reviewed challenges in the paper. For example, it is not clearly discussed how it addresses uncertainty in performance that may arise from subtle changes in an LLM's prompt or how it addresses the data contamination issue.

**Support:**

3

---

> ### Author Rebuttal · Authors · 2025-03-27
>
> Thank you for your recognition and feedback. Below, we try to address your concerns point by point.
>
> > **Q1**: No alternative view/position is presented in the paper?
>
> **A1**: We would like to point out that we include Alternative Views in line 437, page 8. We include a dedicated part titled “Alternative Views”, which states: *"Alternative Views: Adaptive testing research began in the mid-20th century and has developed over the past….."* This section is located near the end of the page and may be easy to overlook.
>
> > **Q2**: It is not clearly discussed how it addresses uncertainty in performance that may arise from subtle changes in an LLM's prompt or how it addresses the data contamination issue?
>
> **A2**: We appreciate this concern and would like to clarify that our paper does propose several feasible ideas to address these challenges:
> 1. Uncertainty in performance due to subtle changes: As discussed in Section 2.1, we draw an analogy to human assessments, where external factors (e.g., fatigue, emotions, environment) can influence observed responses. We propose using Bayesian estimation of ability scores, which naturally includes an uncertainty measure. This allows us to quantify how much prompt variations affect the model’s estimated ability. The paper states:*"The trait being assessed does not change during a short testing period where no new knowledge can be learned, even as observed responses fluctuate."* This means that even if the model's responses change due to prompt variations, our Bayesian framework accounts for this uncertainty, leading to a more robust performance estimation.
> 2. Data contamination issue: In Section 2.2, we outline several ideas inspired by psychometric anomaly detection, including analyzing abnormal response patterns, and examining item characteristics for signs of contamination. Additionally, in Appendix A.3, we conduct a preliminary experiment on MATH, NarrativeQA, and RAFT datasets (Figure 7). The results show that data contamination can be detected using item response characteristics, supporting our proposed approach.
>
> However, we acknowledge that, as a position paper, it only proposes new ideas instead of presenting large-scale experimental validation. Each of these directions represents a largely independent line of work and poses significant challenges that warrant further study. This is precisely why we wrote this position—to encourage further exploration in these areas.
>
> > **Q3**: Is "item characteristic annotation" aspect of the proposed framework used in picking the next question for evaluation at any given point?
>
> **A3**: Yes, item characteristics (e.g., difficulty and discrimination) are directly used for dynamic evaluation. For example, the estimated difficulty/discrimination can be used to select items that either best match the model’s ability or provide the highest discrimination. Thank you for your feedback! We will clarify this further in Section 3.2 to explicitly link item characteristics with dynamic question selection.
>
> > **Q4**: They end the subsection by mentioning *"Recently, Kipnis et al. (2024) apply the Fisher method to identify the most informative items ... they demonstrate that as little as 3% (or even fewer) of the items could be selected to distill a sparse benchmark"* which looks more relevant to item characteristics annotation rather than dynamic item selection. Can the authors clarify?
>
> **A4**: To elaborate, in identifying the "most informative" items, most informative is conditioned on test-taker's ability $\theta$. In other words, an item that provides information for a low-ability model may not be informative for a high-ability one. In practical applications (AI evaluation where model abilities can vary widely), it is common to compute item information at several representative points along the $\theta$ scale (typically from -3 to +3 on the standard normal metric, e.g., at $\theta = -2, -1, 0, 1, 2$). Kipnis et al. (2024) use this method to select and assemble a form provides good information all along the ability continuum. This aligns with the principles of adaptive evaluation in the sense that item informativeness is evaluated w.r.t. ability. But fully adaptive testing can significantly improve on this by identifying items that have maximum information at the current ability estimate [1]. Our original phrasing may be unclear. In the next version, we will explicitly incorporate our discussion and highlight this in Section 3.2. This should help prevent potential misunderstandings.
>
>
> Thank you again for your feedback. If anything unclear, we would clarify further and look forward to continued discussion with you!
>
> Reference:
>
> [1] van der Linden, W. J., (2016). Optimal test assembly. In S. Lane, M. R. Raymond, & T. M. Haladyna (eds.), Handbook of Test Development. Routledge.

---

> > ### Comment · Reviewer_xfA5 · 2025-04-05
> >
> > I'd like to thank the authors for their explanations and answering my questions. I stick with my overall "Accept" recommendation.

---

> > > ### Author Response · Authors · 2025-04-06
> > >
> > > Your encouraging feedback means a lot to us. Thank you again for recognizing the significance of our work, and it motivates us to advance this direction to improve AI evaluation.

---

### Official Review · Reviewer_Z8ZW · 2025-03-13

**Significance:** 4
**Argument Clarity:** 3
**Rating:** 3
**Confidence:** 4

**Questions:**

Could you describe a concrete example for operationalizing psychometric techniques for existing AI evaluation? (Edit: this was addressed in Appendix B)

**Discussion Potential:**

2

**Paper Summary:**

- This paper advocates for adopting ideas from psychometrics in improving AI evaluations because psychometrics focuses on ability-based evaluations and figures out how to weight different evaluation items accordingly.
- In particular, the paper talks about adaptive testing as one way to operationalize AI evaluation.

**Position:**

Yes

**Position In Title:**

Yes

**Related Work:**

2

**Strengths And Weaknesses:**

Strengths:
- Evaluation is a very relevant topic that is important to the ICML community, so offering a new perspective from human testing could promote discussion and new directions of work.
- The position paper is fairly thorough in terms of drawing from external literature on psychometric testing. It is clear the authors have extensive knowledge about the field.

Weaknesses:
- The position paper could provide a clearer background about psychometrics before describing how it might apply to AI evaluation. While the authors are clearly knowledgeable about the area of psychometrics, it may be challenging for a reader to grasp the nuances. Providing a background section for readers would also be helpful.
- Related to the first weakness, the authors attempt to draw connections to adaptive testing as a way to implement psychometric techniques for AI evaluation. However, it is unclear how to operationalize these methods at scale. For example, showcasing more concretely how adaptive testing techniques can apply to an example dataset would be fruitful and demonstrative.

**Support:**

2

---

> ### Author Rebuttal · Authors · 2025-03-27
>
> Thank you for your thoughtful feedback. We appreciate your suggestions for improving our position paper. Below, we address the two weaknesses you highlighted:
>
> > **Q1**: The position paper could provide a clearer background about psychometrics before describing how it might apply to AI evaluation. Providing a background section for readers would also be helpful?
>
> **A1**: Yes. In Section 2.1 (line 92–144), we currently include a brief historical overview of psychometrics and introduced IRT—one of the most fundamental models in psychometrics. Simply put, IRT is most commonly a logistic regression model (infrequently other link functions are used [1]) where the probability of a correct response increases as the test-taker’s latent ability surpasses the item’s difficulty. This core idea is intuitive and widely applicable. Thanks for your feedback, this content might be more visible if moved into a dedicated background section, ensure it isn’t overlooked by readers.
>
> Throughout the paper, we have aimed to explain psychometric concepts using accessible language and concrete examples, rather than delving into abstract technical formulations. For example, the main test only has one equation about psychometrics (a single logistic regression), most of this position focuses on intuitive illustrations. As AI researchers, our goal is to make these ideas approachable for the broader community. Thank you for your suggestion! If any part of our explanation remains unclear, or if you’re interested in further aspects of psychometrics, we'd be delighted to discuss them with you.
>
>
> > **Q2**: It is unclear how to operationalize these methods at scale. For example, showcasing more concretely how adaptive testing techniques can apply to an example dataset would be fruitful and demonstrative. Could you describe a concrete example for operationalizing psychometric techniques for existing AI evaluation?
>
> **A2**: Sure. Although this is a position paper, we not only introduce key ideas and conceptual frameworks, but also provide concrete, practical examples in Appendix B.
>
> In Appendix B: *"B. Case Study: A Simple Implementation of Adaptive Testing for AI Models"*, where we present a detailed example of how adaptive testing techniques can be operationalized for large-scale AI evaluation. This case study outlines the full process, including:
> 1) Estimating item characteristics in the benchmark. It often uses statistical methods to determine properties, e.g., difficulty and discrimination.
> 2) Interactive testing, where an item selection strategy is combined with a psychometric model (user model) to dynamically evaluate AI models through an iterative process: Item selection → Ability estimation update → Next item selection…
> 3) Final ability estimation and result interpretation.
>
> This process closely mirrors how adaptive testing is used in human exams like the GRE, making it a natural extension to AI evaluation. Additionally, we have provided some experimental results and an open-source GitHub repository (anonymous) to facilitate practical implementation (Appendix page 17). Thanks for your suggestion! To improve clarity, we will explicitly reference Appendix B and highlight it in the main text, to ensure readers are aware of this concrete example.
>
> We sincerely appreciate your recognition of our work’s potential impact. If you have any further concerns or suggestions, we would be glad to address them and discuss them with you in more detail.
>
> Reference:
>
> [1] Classe, F. L., & Steyer, R. (2024). A probit multistate IRT model with latent item effect variables for graded responses. European Journal of Psychological Assessment, 40(3), 172–183.

---

### Official Review · Reviewer_HAPz · 2025-03-14

**Significance:** 3
**Argument Clarity:** 3
**Rating:** 3
**Confidence:** 4

**Questions:**

See weakness.

**Discussion Potential:**

3

**Paper Summary:**

This paper presents the position on that we should evaluate AI models using the long standing human testing psychometric principles such as item response theory.
The arguement is made that leveraging these psychometrics, the testing can be made more scientific and robust to outliers and can be used to identify data contamination.
The paper calls for collaborative efforts from the community to adopting this new paradigm of adative testing.

**Position:**

Yes

**Position In Title:**

Yes

**Related Work:**

3

**Strengths And Weaknesses:**

Strength:

The stated position is well supported by the recent research on adaptive testing of AI models using psychometrics. This position is clear contrast with the alternative view of standard model evaluation of simply running the model on every sample.

Weakness:

1. It seems to me that how to use these psychometrics for AI model evaluation is still an ongoning investigation from the works referenced from this position, I wonder if it would be possible for the community to have a widely accepted method for evaluating the models with these psychometrics.
2. There are even more complexed benchmarks introduced and will be introduced in the future such as MLEBench, MLGym, etc which requires a complete agent system to perform the task. This position focuses on standard benchmarks for testing LLMs, however as agentic system become more capable, how can we use these psychometrics to evaluate agents? Especially when the benchmarks for agents often have much less tasks than standard benchmarks, would this lead to unreliable scores?


Overall I rate this paper a weak accept and I hope to see the author's response to my concerns

**Support:**

3

---

> ### Author Rebuttal · Authors · 2025-03-26
>
> We greatly appreciate your thoughtful insights. Your comment on the broader applicability aligns well with our intentions.
>
> > **Q1**: It seems to me that how to use these psychometrics for AI model evaluation is still an ongoning investigation, I wonder if it would be possible for the community to have a widely accepted method for evaluating the models with these psychometrics?
>
> **A1**: Yes, as mentioned in this position, applying psychometrics to AI evaluation is an ongoing research. Recent studies have shown its potential and are gradually gaining acceptance. For example, TinyBench (Polo et al., 2024) and MetaBench (Kipnis et al., 2024) use psychometric techniques to create a ''tiny'' version of various popular benchmarks like MMLU, AlpacaEval, GSM8K, HellaSwag, and TruthfulQA—selecting only 100 items even smaller while still accurately recovering the full benchmark’s scores.
>
> In this position, to make adaptive testing more practical and widely accepted, several promising directions—many of which are already being explored—can help:
>
> 1. Benchmark quality analysis. By collecting model response data from leaderboards, we can estimate the characteristics of each test item. This allows us to detect contaminated and low-quality items, improving the reliability. For example, Rodriguez et al. (2021) successfully used psychometric analysis to automatically identify potential annotation errors in the SQuAD dataset.
> 2. Efficient evaluations. Identifying the most representative / informative items to reduce redundancy and create “tiny” versions. This is especially useful for evaluating AI-generated content, human preferences, or tasks without clear-cut answers, where human annotation is needed, especially in industry. It can lower costs and speed up evaluation cycles.
> 3. Recognizing the limitations of current paradigms, and advocating for further research. While adaptive testing has been widely used in human assessments for over a century, its adoption in AI evaluation is still in its early stages. But awareness is growing and more researchers are recognizing its benefits: By learning from well-established human assessment methods, we can continue improving how we evaluate advanced smart AI models.
>
> > **Q2**: This position focuses on standard benchmarks, how can we use these psychometrics to evaluate agents? Would this lead to unreliable scores?
>
> **A2**: In my view, psychometrics—originally developed to evaluate complex human behavior—can and should be explored to evaluate AI agents. Traditional evaluations often assume that all tasks contribute equally to the final score, and traditional metrics (accuracy/F1/recall) may not be as reliable or informative as they seem. For example, in MLEBench, which consists of 75 tasks, if GPT-4o achieves 5.0% Gold rate while Claude-3.5 achieves 4.4%, we might ask: Is the difference really that small? Could the tasks be too easy overall? Are some tasks bad at distinguishing different models? How to assess an agent’s fine-grained and multidimensional capabilities?
>
> Traditional benchmarking methods cannot answer these questions—but psychometrics can, because it:
> 1. Recognizes that not all tasks are equally important and accounts for differences in difficulty and discriminative power.
> 2. Models latent ability rather than relying on raw scores, providing valuable insights into confidence, fairness, and interpretability (Section 2.1).
> 3. Enables comparability across different benchmarks by taking an ability-oriented approach. It focuses on the characteristics of the test items rather than the items themselves (see Page 4 for details). This is similar to how GRE scores are scaled, allowing for fair comparisons across different test versions.
> 4. Supports fine-grained, multidimensional skill modeling. Using psychometrics and cognitive diagnostic models, we can consider complex relationships between knowledge components and skills. This enables a multidimensional "capability radar" that represents the nuanced abilities of AI agents (see Section 5.1 future opportunities).
>
> Agentic benchmarks often contain fewer tasks than standard LLM benchmarks, making traditional score (e.g., Accuracy) reliability a concern. Luckily, psychometric methods do not rely on its large number of tasks/items, making them a more precise and reliable way to evaluate AI agents.
>
> This is a great topic for our further research, and I’ll expanding Section 5 to highlight advantages of using adaptive testing to evaluate complex AI agent tasks. Looking forward to more discussions with you on this!
>
> Reference:
>
> [1] Polo F M, Weber L, Choshen L, et al. tinyBenchmarks: evaluating LLMs with fewer examples ICML, 2024.
>
> [2] Kipnis A, Voudouris K, Buschoff L M S, et al. metabench-A Sparse Benchmark to Measure General Ability in Large Language Models. arXiv preprint, 2024.
>
> [3] Rodriguez P, Barrow J, Hoyle A M, et al. Evaluation examples are not equally informative: How should that change NLP leaderboards? ACL-IJCNLP. 2021.

---

### Official Review · Reviewer_rgHd · 2025-03-14

**Significance:** 4
**Argument Clarity:** 4
**Rating:** 4
**Confidence:** 2

**Questions:**

1. How could this work in practice?
2. what changes would we expect to see in the outcome?
3. Which researchers would need to work together to make this happen?
4. How do we overcome the known limiations of adaptive testing in education reseach? (items being known, items space selection etc.)
5. What about differential item funcitoning, is it reasonable to have only one? shouldn''t the model allow for subgroups, how would that be organized?

**Discussion Potential:**

3

**Paper Summary:**

This is a position paper that clearly outlines how humans are tested in the educational setting (adaptive testing) and explains the underlying measurement theory (latent trait/variable analysis). The psychometrics field has developed these methods for years and this position paper is making a strong claim that a transfer of knowledge from that area to ML is necessary.

**Position:**

Yes

**Position In Title:**

Yes

**Related Work:**

3

**Strengths And Weaknesses:**

The paper is well outlined and well structured. The suggestions made are grounded in mathematical theory.

A few places for improvement. it would be nice to seeempirical validation. how would this work in practice? While the paper points to the potential advantages of adaptive testing, it does not provide large-scale experiments or concrete case studies demonstrating its superiority over existing AI benchmarks. This could be ok for a position paper, however the argument is stronger with such pointers.

Additionally, the proposed paradigm shift requires significant infrastructure changes, which may hinder adoption in the AI research community, where static benchmarking is deeply ingrained.

There is also a risk that adaptive testing could introduce new biases—for example, if item selection does not account for model-specific weaknesses or variations, it might inadvertently favor certain architectures or training methodologies.

**Support:**

3

---

> ### Author Rebuttal · Authors · 2025-03-26
>
> We sincerely appreciate your recognition of our position and its potential impact on AI community. Below, we address your questions point by point:
> > **Q1**: How could this work in practice?
>
> **A1**: As shown in Section 2.1, recent works have already begun to demonstrate our position's feasibility, i.e., integrating psychometrics into AI evaluation and provide large-scale experiments. For example, TinyBench (Polo et al., 2024) utilizes psychometrics to construct tiny versions (only 100 samples) of various benchmarks e.g., MMLU, HELM, and AlpacaEval. Similarly, MetaBench (Kipnis et al., 2024) applies adaptive testing to identify the 3% most informative items across various benchmarks (GSM8K, HellaSwag, TruthfulQA, etc). While this is a position paper, we share your concerns regarding its applicability. Thus, we provide a scalable framework and illustrative experiments in Appendix B, and we also share a practical repository (anonymous github in page 17).
>
> > **Q2**: What changes would we expect to see in the outcome?
>
> **A2**: We expect to see reliable evaluations, especially for models affected by data contamination or low-quality benchmark. Additionally, benchmark size required for evaluation should decrease significantly, which is particularly valuable in industry settings where large-scale human annotation is expensive. It not only reduces evaluation costs but also improves the robustness of AI benchmarking.
>
> > **Q3**: Which researchers would need to work together to make this happen?
>
> **A3**: Shifting away from static benchmarking is challenging because: it has been the standard in our community for decades. Our AI researchers alone may not be able to drive this change, as traditional evaluations are deeply embedded in research practices and educational materials. Collaboration is needed between psychometricians, benchmark designers, and software engineers. For example, psychometricians can refine item selection strategies, and AI evaluation experts can ensure compatibility with existing benchmarks. Even if this shift happens gradually, raising awareness is crucial: Facing the so-called AGI of increasing intelligence like human, the traditional evaluation paradigm may no longer be sufficient.
>
> > **Q4**: How do we overcome the known limitations of adaptive testing in education research?
>
> **A4**: Adaptive testing in human assessment has evolved over more than a century, addressing many challenges that now also appear in AI evaluation (e.g., item exposure, unexpected behavior, item biases, and automated item generation). Established solutions in psychometrics, including exposure control algorithms (e.g., randomization, stratification), help prevent overfitting and data leakage; IRT is widely used to detect item bias, ensuring fairness across groups... We believe that these solutions are highly relevant to AI benchmarking, where similar issues are increasingly problematic. This is precisely why we advocate for the transfer and adaptation of psychometric methodologies to the AI domain.
>
> > **Q5**: What about DIF and biases, is it reasonable to have only one? shouldn''t the model allow for subgroups? favor certain architectures or training methodologies?
>
> **A5**: This is a great question. Both human and AI assessments face the challenge of certain items favoring specific groups. DIF is a well-studied issue in psychometrics, where statistical methods are used to detect and mitigate biases in test items (Camilli et al., 1994; Hambleton et al., 1991). The key question is: what is the goal of the evaluation? If the objective is specifically to assess a student's reasoning ability within the context of board games, then naturally, those unfamiliar with chess would be at a disadvantage. Similarly, if we are evaluating AI's reasoning ability, retrieval-based models may be at a disadvantage. Of course, it can offer tools for multi-dimensional diagnostics, where models can be evaluated across fine-grained skills or knowledge concepts (page 3). But the best approach is to carefully design benchmarks so that they align with the intended evaluation goals. Psychometric methods provide systematic ways to identify and remove such items, ensuring that benchmarks remain fair. We welcome further discussion about this.
>
>
> We truly appreciate your insights and look forward to further refining our work based on this valuable feedback.
>
> Reference:
>
> [1] F.M. Polo, L. Weber, L. Choshen, Y. Sun, G. Xu, and M. Yurochkin, “tinybenchmarks: evaluating llms with fewer examples,” in Forty-first International Conference on Machine Learning, 2024.
>
> [2] A. Kipnis, K. Voudouris, L. M. S. Buschoff, and E. Schulz, “metabench–a sparse benchmark to measure general ability in large language models,” arXiv preprint arXiv:2407.12844, 2024.
>
> [3] G. Camilli and L. A. Shepard, Methods for identifying biased test items, vol. 4. Sage, 1994.
>
> [4] R. K. Hambleton, H. Swaminathan, and H. J. Rogers, Fundamentals of item response theory, vol. 2. Sage, 1991.

---

> > ### Comment · Reviewer_rgHd · 2025-04-07
> >
> > Thanks for your commnts. I will keep my  "Accept" recommendation.

---

> > > ### Author Response · Authors · 2025-04-08
> > >
> > > Thank you for recognizing the necessity of our work for the ML community. Your positive feedback is encouraging and motivates us to further explore this direction.

---

### Decision · Program_Chairs · 2025-04-30

**Decision:**

Accept (poster)

**Comment:**

The reviews are consistent in viewing that the perspective of using IRT to better understand the results of leaderboard evaluations, to discover problematic test items, and to provide more interpretable evaluations is valuable. The weaknesses of the paper are relatively minor and likely addressable in a revision. The authors are encouraged to provide more information on how IRT can inform dynamic leaderboards in either item selection, construction of examples (e.g., adversarial examples), or on better comparisons between computer and human ability. Based on the rebuttals and author discussion, this seemed where most of the questions were.